# Presence of *Pseudomonas aeruginosa* in feces exacerbate leaky gut in mice with low dose dextran sulfate solution, impacts of specific bacteria

Wimonrat Panpetch[1,2], Somying Tumwasorn[1]*, Asada Leelahavanichkul[1,3]*

1 Department of Microbiology, Faculty of Medicine, Chulalongkorn University, Bangkok, Thailand,
2 Department of Microbiology, Faculty of Science, Burapha University, Chonburi, Thailand, 3 Center of Excellence in Translational Research on Immunology and Immune-Mediated Diseases (CETRII), Department of Microbiology, Faculty of Medicine, Bangkok, Thailand

☯ These authors contributed equally to this work.
* aleelahavanit@gmail.com (AL); somying.T@chula.ac.th (ST)

## Abstract

The impact of *Pseudomonas aeruginosa* (PA) was explored in a mouse model with non-diarrheal gut permeability defect using 1.5% dextran sulfate solution (DSS) plus antibiotics (ATB) with or without orally administered PA. As such, ATB+DSS+PA mice induced more severe intestinal injury as indicated by stool consistency and leaky gut (FITC-dextran assay, bacteremia, and endotoxemia) with an increase in serum cytokines, liver enzyme, and hepatocyte apoptosis when compared with ATB+DSS mice. There was no abnormality by these parameters in the non-DSS group, including water alone (control), antibiotics alone (ATB +water), and antibiotics with PA (ATB+water+PA). Despite a similarly fecal microbiome patterns between ATB+DSS and ATB+DSS+PA groups, a higher abundance of *Pseudomonas*, *Enterococci*, and *Escherichia-Shigella* was detected in ATB+DSS+PA mice. Additionally, the additive pro-inflammation between pathogen molecules, using heat-killed *P. aeruginosa* preparations, and LPS against enterocytes (Caco2) and hepatocytes (HegG2), as indicated by supernatant IL-8 and expression of several genes (*IL-8*, *NF-kB*, and *NOS2*) are demonstrated. In conclusion, presence of *P. aeruginosa* in the gut exacerbated DSS-induced intestinal injury with spontaneous translocation of LPS and bacteria from the gut into the blood circulation (leaky gut) that induced more severe systemic inflammation. The presence of pathogenic bacteria, especially PA in stool of the healthy individuals might have some adverse effect. More studies are in needed.

## Introduction

Determination of fecal microbiome in healthy hosts has been currently performed as a health prediction biomarker [1, 2]. As such, Firmicutes (mostly Gram-positive anaerobes) and Bacteroides (mostly Gram-negative anaerobes) are predominant in the gut of healthy hosts with

**Data Availability Statement:** Data are available from: https://github.com/wimonratpan/minidataset. Nucleic acid sequences in this study were

deposited to an open access Sequence Read Archive (SRA) database of NCBI, the accession number of the dataset PRJNA1012512 and the DOI of the data is https://www.ncbi.nlm.nih.gov/sra/?term=PRJNA1012512.

**Funding:** This research is supported by the Program Management Unit for Human Resources & Institutional Development, Research and Innovation (B16F640175 and B48G660112) with Rachadapisek Sompote Matching Fund (RA-MF-22/65, RA-MF-13/66, and RA-MF-eAsia), and Thailand Science Research and Innovation Fund Chulalongkorn University (HEAF67300087) as well as National Research Council of Thailand (NRCT-N41A640076 and NRCT-N34A660583). WP was supported by Rachadapisek Sompote Fund for Postdoctoral Fellowship, Chulalongkorn University. The funders had no role in study design, data collection and analysis, decision to publish, or preparation of the manuscript.

**Competing interests:** The authors declare no conflict of interest. The authors declare that the research was conducted in the absence of any commercial or financial relationships that could be construed as a potential conflict of interest.

some age-related characteristics [3]. Interestingly, healthy hosts have an increased abundance of pathogenic bacteria, especially Gram-negative aerobes from the phylum Proteobacteria, which might be associated with asymptomatic dysbiosis [4, 5] but are prone to intestinal injury. However, the influence and management of predominant Proteobacteria in feces of the healthy individuals is still in question. Here, a mouse model using low-dose dextran sulfate solution (DSS) and *Pseudomonas aeruginosa* (the representative Proteobacteria) were used to test the hypothesis. Accordingly, *P. aeruginosa* are important bacteria with high pathogenicity [6], high incidence of antibiotic resistance [7], and a great ability to thrive on the moist surfaces of medical equipment [8]. As a Gram-negative bacterium, the lipopolysaccharide (LPS) of *P. aeruginosa* triggers inflammatory cascades [9], facilitates biofilm formation [10], and induces sepsis (a syndrome of the imbalance responses to pathogens) [11]. Meanwhile, the outer membranes (OMPs) of *P. aeruginosa* facilitate nutrient exchange, adhesion, and antibiotic tolerance [10]. Discovery of *P. aeruginosa* in feces indicates gut dysbiosis (an imbalance of gut microbiota), which is due in part to *P. aeruginosa*' s greater ability to tolerate harsh microenvironments than other normal bacteria [12–14]. Increased *P. aeruginosa* in the gut might facilitate the growth of other Gram-negative bacteria and *Candida albicans*, the most common and the second most abundant microorganisms in the human intestines, respectively, that affect the intestinal integrity supporting gut dysbiosis-induced enterocyte injury [15]. In several pathogenic situations, gut dysbiosis causes gut permeability, leading to the translocation of microbial molecules, such as LPS (the major component of Gram-negative bacterial cell walls) and (1→3)-β-D-glucan (BG; the main cell wall component of *C. albicans*), from the gut into the blood circulation (leaky gut), resulting in worsening systemic inflammation [16–18]. Indeed, several insults can cause gut barrier defects, including medications [13, 19], systemic infections [17, 18], obesity [20], and strenuous exercise [21]. Additionally, there have also been reports of elevated *P. aeruginosa* in feces of mice suffering from sepsis [22, 23]. Because pathogens in the healthy gut are contained within the gut and separated from the blood circulation by the gut barrier, strengthening the gut barrier might be beneficial in individuals with positive *P. aeruginosa* in their feces even without any symptoms. Due to reports on the high-virulence pathogenic bacteria in healthy individuals [24], the detectable of these bacteria might make the host more susceptible to intestinal injuries. Antibiotic administration and repeated administration of pathogenic bacteria are necessary for the sustained presentation of the administered bacteria in mouse feces [13] supporting the role of healthy gut microbiota in the prevention of gut dysbiosis [25]. On the other hand, DSS, a sulfated polysaccharide that interferes with enterocyte tight junctions, is a well-known substance used for studying leaky gut as a high dose of DSS (3%) induces overt colitis model with diarrhea [26–29], while a low dose DSS (1–1.5%) causes leaky gut without diarrhea [30–32]. Hence, we tested if an increased abundance of *P. aeruginosa* in healthy mice can worsen leaky gut, using low-dose dextran sulfate solution (DSS) with antibiotics following a published protocol [13].

## Materials and methods

### Animals and animal models

Animal procedures were carried out in accordance with the US National Institutes of Health guidelines and followed the 8th Edition of the Guide for Care and Use of Experimental Animals, published by the National Research Council of the National Academies (2011; available at https://grants.nih.gov/grants/olaw/guide-for-the-care-and-use-of-laboratory-animals.pdf) and the Animal Research: Reporting of In Vivo Experiments (ARRIVE) guidelines. Also, the study is reported in accordance with ARRIVE guidelines. The experimental procedure was approved by the Institutional Animal Care and Use Committee of the Faculty of Medicine,

Chulalongkorn University, Bangkok, Thailand (ASP SST 10/2557). Researchers have been trained in the scientific use of animals and received certification from the National Research Council of Thailand (NRCT). Male C57BL/6 mice, aged 8 weeks, were purchased from the Nomura Siam International Co., Ltd. (Lumphini, Pathumwan, Bangkok, Thailand). All mice were kept in an animal facility designed for the use of bacteria in mice, with a light-dark cycle 12:12 at the Animal Center Faculty of Medicine, Chulalongkorn University. Mice were housed in the same cage for the assessment of several parameters of *Pseudomonas aeruginosa*-infected mice in the ATB+DSS-administered model, whereas some mice in each experimental group were kept in separate cages for the microbiota analysis. To compare the data using one-way analysis of variance (ANOVA) between 5 independent groups, we use the G-power tool to calculate the sample size per group [33]. As a result, the suitable statistical test was obtained using the smallest number of mice (7 mice per group), and significance was determined using *p*-value < 0.05. To investigate the effect of *Pseudomonas aeruginosa* in an ATB+DSS administered mouse model, mice were randomly divided into five groups: the control group (n = 5), the ATB+water group (n = 5), the ATB+water+PA (*P. aeruginosa*) group (n = 5), the ATB+DSS administered group without PA (n = 7), and the ATB+DSS administered group with PA (n = 7).

Then, the antibiotic cocktail (ATB) and dextran sulfate sodium (DSS) were used to eliminate gut microbiota and induce gut barrier damage, respectively (Fig 1A). To induce non-diarrheal leaky gut, 1.5% (w/v) DSS (Sigma-Aldrich, St. Louis, MO, USA) or regular drinking water was used for 11 days of the experiments. After 1-day post-DSS, oral administration of antibiotics cocktail (Sigma-Aldrich, St. Louis, MO, USA), containing gentamicin (3.5 mg/kg), colistin (4.2 mg/kg), metronidazole (21.5 mg/kg) and vancomycin (4.5 mg/kg), at 100 μL/ dose twice a day for 3 days followed by an intraperitoneal injection of a single dose of clindamycin (10 mg/kg) [34] at the 5th day of experiments before the oral administration of 0.3 mL *P. aeruginosa* ($1\times10^9$ CFU) for another 5 days. Notably, *Pseudomonas aeruginosa*, a representative pathogenic bacterium, from the American Type Culture Collection (ATCC) number 27853 (Manassas, VA, USA), was cultured on Tryptic soy agar (TSA) (Oxoid Ltd., Hampshire, UK) at 37°C for 24 h under aerobic conditions before being quantitatively prepared with an optical density of 600 nanometers (OD600) in 0.3 mL phosphate buffer solution (PBS). The mice were observed daily and their stool consistency was semi-quantitatively evaluated according to the following score; 0, normal; 1, soft or loose; 2, diarrhea, as previously published [35]. Humane endpoints were utilized to euthanize anxious mice. The criteria for euthanasia were inability to maintain an upright position, hunched posture, agonal breathing, weight loss of more than 15%, and reduced mobility following stimulation. Mice were monitored on a daily basis, and mice that met the criteria were euthanized via cardiac puncture under isoflurane anesthesia. On the 11th day of the experiments (1 day after the last dose of *P. aeruginosa*), mice were sacrificed with cardiac puncture under isoflurane anesthesia with sample collection. For histology and tissue cytokines, the organs, including the ileum (proximal to caecum), caecum, colon (distal to caecum), livers, and kidneys, were stored in 10% neutral formalin or snap-frozen in liquid nitrogen (then stored at -80°C), respectively.

## Gut permeability determination

The fluorescein isothiocyanate-dextran dextran (FITC-dextran) assay and the spontaneous detection of bacteria (culture) or lipopolysaccharide (LPS or endotoxin) in blood were used to measure gut permeability [20, 30, 36]. The detection of viable bacteria or microbial compounds in blood without active infection, or the presence of FITC-dextran (a non-intestinally absorbable carbohydrate) in serum following an oral gavage, suggest a deficiency in gut

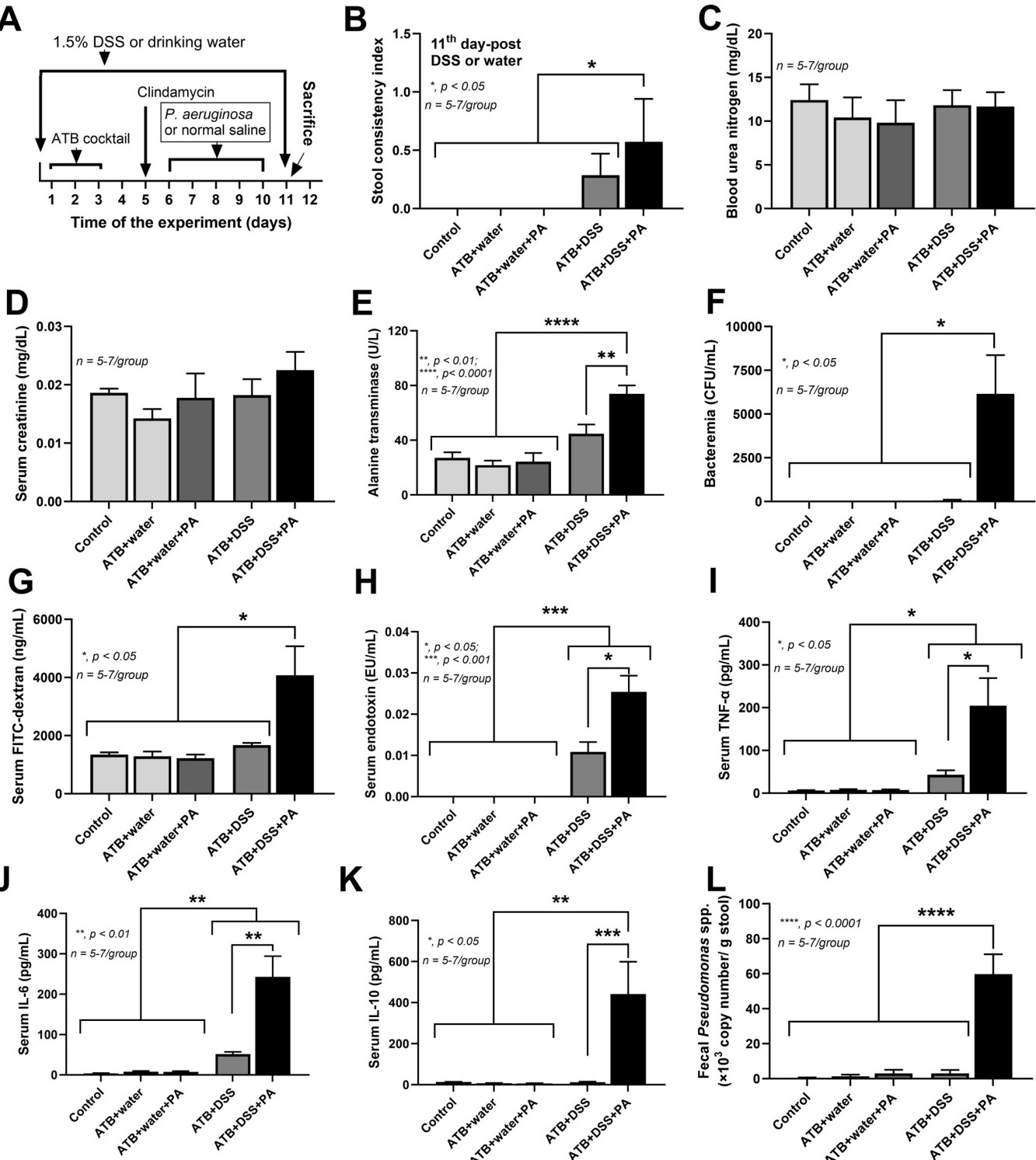

**Fig 1. *P. aeruginosa* administration exacerbated leaky gut in DSS mice with antibiotics.** A schematic representation of the study protocol using dextran sulfate solution (DSS), antibiotics (ATB), and *Pseudomonas aeruginosa* is demonstrated (A). Characteristics of mice in the non-DSS groups, including water control (control), antibiotics without or with *P. aeruginosa* (ATB+water and ATB+water+PA), and the DSS groups, including dextran sulfate solution (DSS) without or with *P. aeruginosa* (ATB+DSS and ATB+DSS+PA) as indicated by stool consistency index (B), renal function with blood urea nitrogen and serum creatinine (C, D), liver enzyme (alanine transaminase) (E), leaky gut parameters (bacteremia, FITC-dextran assay, serum endotoxemia) (F-H), serum cytokines (TNF-α, IL-6, and IL-10) (I-K), and the abundance of *Pseudomonas* spp. in mouse feces (L) are demonstrated (n = 5–7/group). Significant differences *, $p < 0.05$; **, $p < 0.01$; ***, $p < 0.001$; ****, $p < 0.0001$ were compared between the indicated groups.

permeability [13, 37]. For the procedure of FITC-dextran assay, 25 mg/mL (0.5 mL) FITC-dextran (Sigma-Aldrich) was orally administered for 3 h prior to sacrifice, and the serum FITC-dextran was then quantified using fluorospectrometry (Thermo Scientific, Wilmington, DE, USA) with the excitation and emission wavelengths at 485 and 523 nm, respectively, with a serially diluted FITC-dextran standard curve [13, 37]. For bacteremia, blood (25 μL) was directly spread onto blood agar plates (Oxoid, Germany) and incubated at 37°C for 24 h before counting bacterial colonies [30]. For LPS detection, serum samples were processed by HEK-Blue LPS detection (InvivoGen, San Diego, CA, USA).

## Serum, histological, and fecal analyses

An enzyme-linked immunosorbent assay (ELISA; Invitrogen, Carlsbad, CA, USA) was used to measure the levels of cytokines, including tumor necrosis factor (TNF)-α, interleukin (IL)-6, and IL-10, in serum and in homogenized tissue. For homogenized tissue samples, the organs were weighed and homogenized by the Ultra-Turrax homogenizer (IKA, Staufen, Germany) in 500 μL PBS (pH 7.4) with protease inhibitors and centrifuged at 12,000 × g for 15 min at 4°C to collect the supernatant. Tissue inflammation was represented by the cytokines in the supernatant [14, 19]. For histology, the organs were preserved in 10% formalin, embedded in paraffin, sliced to a 4 μm thickness, and stained with Hematoxylin-Eosin (H&E) color. The liver injury was assessed blindly by two pathologists in 10 randomly selected fields at 200 magnification for each animal, based on cell congestion, cellular degeneration, cytoplasmic vacuolization, leukocyte infiltration, and cellular necrosis with the following damage area score per the examined field: 0 area less than 10%; 1 damage 10–25%; 2 damage involving 25–50%; 3 damage 50–75%, and 4 indicates 75–100% of the area being affected, following previous publications with modification [20, 38]. In parallel, immunohistochemistry with an anti-active caspase-3 antibody (Cell Signaling Technology, MA, USA) was used to assess hepatic apoptosis.

The apoptotic cells on each slide were counted at 200× magnification and presented as the number of positive cells per high-power field as previously mentioned [20, 30]. To determine an abundance of *Pseudomonas* spp. in feces, the quantitative real-time PCR was performed. The total DNAs were extracted using a QIAamp fast DNA Stool Mini Kit (Qiagen, Hilden, Germany) and amplified in accordance with manufacturer's instructions using the primers for variable regions of 16S rRNA gene sequence of *Pseudomonas* genus-specific primer; PA (forward; 5′-GACGGGTGAGTAATGCCTA-3′) and (reverse; 5′-CACTGGTGTTCCTTCCTATA-3′) [39]. The amplicon was approximately 618 base pairs (bp), and *Pseudomonas aeruginosa* (also designated as PA ATCC 27853) had a genomic size of 6,833,187 bp [40]. The bacterial genome is approximately $4.51 \times 10^9$ g/mol and contains $6.02 \times 10^{23}$ molecules/mol. One bacterium corresponds to 7.5 fg of DNA. The constructive standard curve was generated by the QuantStudio™ Design & Analysis Software v1.4.3 (Thermo Fisher Scientific) using 10-fold serial dilution (15 fg to 1,500 pg) with bacterial concentrations ranging from 2 to $2 \times 10^5$ bacteria [33].

For histology in colon tissues, 10% formalin fixed paraffin-embedded colon sections were stained with Hematoxylin and Eosin (H&E) (Sigma-Aldrich) and examined by two pathologists in a blinded manner for intestinal damage with the modified semi-quantitative score at 200× magnification according to previous publication [33]. The criteria included mononuclear cell infiltration (in mucosa and sub-mucosa), epithelial hyperplasia (epithelial cell in longitudinal crypts), reduction of goblet cells, and epithelial cell vacuolization in comparison with control following scores; 0; leukocyte < 5% and no epithelial hyperplasia (<10% of control), 1; leukocyte infiltration 5–10% or hyperplasia 10–25%, 2; leukocyte infiltration 10–25% or hyperplasia 25–50% or reduced goblet cells (>25% of control), 3; leukocyte infiltration 25–50% or hyperplasia >50% or intestinal vacuolization, 4; leukocyte infiltration >50% or ulceration.

## The gene expression of tight junction proteins in colon of the mice using qRT-PCR

To measure the intestinal tight junction damage, gene expression of occludin and ZO-1 in control, ATB+DSS+PA, ATB+DSS groups were evaluated by quantitative reverse transcription polymerase chain reaction (qRT-PCR) as described int the published protocol [33]. Shortly, colon tissue samples were treated with an RNA-easy mini kit (Qiagen, Hilden, Germany) to extract total RNA, which was then measured using a Nanodrop 1000 Spectrophotometer (Thermo Scientific, Wilmington, DE, USA). The complementary DNA (cDNA) was performed using a High-Capacity cDNA Reverse Transcription (Applied Biosystems, Warrington, UK) before being subjected to qRT-PCR with SYBR Green PCR Master Mix on a QuantStudio6 Flex Real-time PCR System (Thermo Fisher Scientific). The sequences of primers were as follows: mouse occludin (forward; 5′-CCTCCAATGGCAAAGTGAAT-3′ and reverse; 5′- CTCCCCACCTGTCGTGTAGT-3′), mouse ZO-1 (forward; 5′- GCAAGAGGAGTCCCTGACT G-3′ and reverse; 5′-CGGCTCTGTCCTAACTCCAG-3′), and the endogenous housekeeping gene *β-actin* (forward; 5'-CGGTTCCGATGCCCTGAGGCTCTT-3' and reverse; 5'-CGTCACA CTTCATGATGGAATTGA-3'). The results were expressed in terms of relative quantification with *β-actin* to standardize the comparative threshold (delta-delta Ct) method ($2^{-\Delta\Delta Ct}$).

## Fecal microbiome analysis

Feces from each mouse (0.25 g per mouse; 3 mice per group) from different cages in each experimental group were collected for the microbiota analysis. Total metagenomic DNA for prokaryotes was obtained using the QIAamp PowerFecal Pro DNA Kit (Qiagen, Hilden, Germany) according to the manufacturer's protocols. Briefly, 0.25 g of fecal samples were extracted, and the collected DNA was measured using a DeNovix QFX Fluorometer. The prokaryotic 16S rRNA gene at the V3V4 region was performed using the Qiagen QIAseq 16S/ITS Region panel (Qiagen, Hilden, Germany) and amplicons were labeled with different sequencing adaptors using QIAseq 16S/ITS Region Panel Sample Index PCR Reaction (Qiagen, Hilden, Germany). The quality and quantity of DNA libraries were evaluated using DeNovix QFX Fluorometer and QIAxcel Advanced (Qiagen, Hilden, Germany), respectively. 16S rRNA libraries were sequenced using an Illumina Miseq600 platform (Illumina, San Diego, CA, USA) and the sequences were processed following DADA2 v1.16.0 pipeline (https://benjjneb. github.io/dada2/). The DADA2 pipeline describes microbial diversity and community architectures of microorganisms using unique amplicon sequence variants (ASVs) [41]. Silva version 138, a reference database, was used to classify microbial taxa [42]. Alpha diversity index (Chao1 richness and Shannon evenness estimation) was computed using DADA2 software. For Bata diversity, non-metric multidimensional scaling (NMDS) based on Bray-Curtis dissimilarity and principal coordinate analysis (PCoA) were plotted from Phyloseq data. A pairwise comparison of alpha diversity (Chao1 and Shannon) was calculated using Tukey's multiple comparisons test ($p < 0.05$). Permutational multivariate analysis of variance (PERMANOVA) was performed to evaluate the significant differences in beta diversity among groups at $p < 0.05$. The 16S rDNA sequences in this study were deposited in an NCBI open access Sequence Read Archive database with accession number PRJNA1012512.

## The *in vitro* experiments

In two cell lines, human colonic epithelial Caco-2 cells (ATCC HTB-37) and human hepatoma HepG2 cells (ATCC HB8065), the inflammatory responses in enterocytes and hepatocytes, respectively, were tested against the Dulbecco's Modified Eagle Medium (DMEM) media

**Table 1. List of human primers in this study.**

| Target | Primer sequence | |
|---|---|---|
| | Forward | Reverse |
| IL-8 [44, 45] | 5′-ACACTGCGCCAACACAGAAATTA-3′ | 5′-TTTGCTTGAAGTTTCACTGGCATC-3′ |
| IL-6 | 5′-ATGAACTCCTTCTCCACAAGC-3′ | 5′-GTTTTCTGCCAGTGCCTCTTTG-3′ |
| IL-10 | 5′-TCTCCGAGATGCCTTCAGCAGA-3′ | 5′-TCAGACAAGGCTTGGCAACCCA-3′ |
| TNF-α | 5′-CTCTTCTGCCTGCTGCACTTTG-3′ | 5′-ATGGGCTACAGGCTTGTCACTC-3′ |
| NF-κB [46] | 5′-ATGGCTTCTATGAGGCTGAG-3′ | 5′-GTTGTTGTTGGTCTGGATGC-3′ |
| TLR2 | 5′- TCCTCCAATCAGGCTTCTCTGTCTT-3′ | 5′- CTCGCAGTTCCAAACATTCC-3′ |
| MUC2 [34, 47] | 5′-CCTGCCGACACCTGCTGCAA-3′ | 5'-ACACCAGTAGAAGGGACAGCACCT-3' |
| Claudin-1 | 5′-AAGTGCTTGGAAGACGATGA-3′ | 5′-CTTGGTGTTGGGTAAGAGGTT-3′ |
| Occludin [48] | 5′-CCAATGTCGAGGAGTGGG-3′ | 5′-CGCTGCTGTAACGAGGCT-3′ |
| ZO-1 | 5′-ATCCCTCAAGGAGCCATTC-3′ | 5′-CACTTGTTTTGCCAGGTTTTA-3′ |
| Muc2 [34, 47] | 5'-CCTGCCGACACCTGCTGCAA-3' | 5′-ACACCAGTAGAAGGGACAGCACCT-3′ |
| bcl-2 [49] | 5′- GGTGCCACCTGTGGTCCACCT-3′ | 5′-CTTCACTTGTGGCCCAGATAGG-3′ |
| casp3 [50] | 5′-CATGGAAGCGAATCAATGGACT-3′ | 5′-CTGTACCAGACCGAGATGTCA-3′ |
| casp8 [50] | 5′-TTTCTGCCTACAGGGTCATGC-3′ | 5′- TGTCCAACTTTCCTTCTCCCA-3′ |
| casp9 [50] | 5′-CTCAGACCAGAGATTCGCAAAC-3′ | 5′-GCATTTCCCCTCAAACTCTCAA-3′ |
| NOS2 [51] | 5′-CAGCGGGATGACTTTCCAAG-3′ | 5′-AGGCAAGATTTGGACCTGCA-3′ |
| GAPDH [44, 45] | 5′-GCACCGTCAAGGCTGAGAAC-3′ | 5′-ATGGTGGTGAAGACGCCAGT-3′ |

IL-8, Interleukin-8; IL-6, Interleukin-6; IL-10, Interleukin-10; TNF-α, Tumor necrosis factor-α; NF-κB, Nuclear factor kappa B; TLR-2, Toll-like receptor 2; MUC2, mucin2; ZO-1, Zonula occludens-1; Casp8, Caspase 8; Casp9, Caspase 9; NOS2, Nitric oxide synthase 2, GAPDH, glyceraldehyde-3-phosphate dehydrogenase

control or molecules of *Pseudomonas* (PA; using heat-killed preparations) or lipopolysaccharide (LPS) or PA plus LPS (PA+LPS). *Pseudomonas aeruginosa* ($1.5 \times 10^9$ cells/ mL) was heated (65°C for 1 h) before being sonicated (power amplitude 38%, pulse on and off for 20 and 5 seconds, respectively, and total processing time 60 min) with a High-Intensity Ultrasonic Processor (VC/VCX 130, 500,750) [19]. According to previous publications, the multiplicity of infection (MOI) for *P. aeruginosa* (the ratio of cells versus organisms) was 1:300 [13, 14, 20]. Notably, Caco-2 and HepG2 cells were maintained in DMEM supplemented with 20% and 10% (v/v) heat-inactivated fetal bovine serum (Gibco-Invitrogen, USA), respectively, at 37°C in a humidified 5% $CO_2$ [30]. Caco-2 or HepG2 at $5 \times 10^4$ cells/well were treated with media alone or media containing with the preparations from *Pseudomonas* (PA), LPS from *Escherichia coli* O26:B6 (Sigma-Aldrich) at 1 μg/mL, or PA+LPS (adjusted into the same volume per well). After incubation, the supernatant was collected by centrifugation ($125 \times g$, 4°C for 7 min) and cytokines were measured by ELISA kit (Invitrogen, Carlsbad, CA, USA), while the gene expression was evaluated by quantitative reverse-transcription polymerase chain reaction (qRT-PCR) as previously described [13, 20]. In short, total RNA was isolated from treated cells using the TRIzol reagent (Invitrogen, USA). A high-capacity reverse transcription assay (Applied Biosystems, Warrington, UK) was used to convert the RNA (50 ng) into complementary DNA (cDNA). Gene expression was measured in relation to GAPDH expression using $2^{-\Delta\Delta C_P}$ method [43] with SYBR Green PCR Master Mix and QuantStudioTM Design & Analysis Software v1.4.3 (Thermo Fisher Scientific) [30]. The list of primers for PCR is presented in Table 1.

## Statistical analysis

The mean ± standard error of the mean (SEM) was used for data presentation. The differences between groups were examined by one-way analysis of variance (ANOVA) followed by Tukey's analysis or Student's *t*-test for comparisons of multiple groups or 2 groups, respectively. All statistical analyses were performed with Graph Pad Prism version 9.0 software (La Jolla, CA, USA). A *p*-value of $< 0.05$ was considered statistically significant.

## Results

### *P. aeruginosa* administration worsened leaky gut in DSS mice with antibiotics

As such, mice with 1.5% DSS or drinking water control were given an antibiotic cocktail (ATB) with/ without daily *P. aeruginosa* (PA) gavage for 5 days (Fig 1A). Without DSS (ATB alone and ATB with PA), the mouse characteristics, including stool consistency, renal function (blood urea nitrogen and serum creatinine), liver enzyme (serum alanine transaminase), leaky gut (bacteremia, FITC-dextran assay, and serum endotoxin), and serum cytokines (TNF-α, IL-6, and IL-10), were not different to the control (Fig 1B–1K), supporting impacts of healthy gut barrier [25]. In comparison with the control, 5 in 7 ATB+DSS mice demonstrated normal stool consistency with systemic inflammation (alanine transaminase, endotoxemia, TNF-α, and IL-6) (Fig 1B–1K), implying the leaky gut without an overt diarrhea [32, 37]. When comparing ATB+DSS with *P. aeruginosa* (ATB+DSS+PA) and ATB+DSS mice, *P. aeruginosa* exacerbated loose stool in almost all mice, leaky gut parameters (bacteremia, FITC-dextran, and endotoxemia), systemic inflammation (TNF-α, IL-6, and IL-10), and liver damage (alanine transaminase). However, there was no kidney injury (Fig 1B–1K) similar to the previous reports [13, 34]. It is noteworthy that the mucosal injury might be required for *Pseudomonas* colonization, since *Pseudomonas* detection using polymerase chain reaction (PCR) was demonstrated only in ATB+DSS+PA mice, but not in the non-DSS groups or ATB+DSS group (Fig 1L). Moreover, the colon length, morphology, and the tight junction proteins gene expression including occludin and ZO-1 in each group of mice were demonstrated in S1 Fig.

Then, the cytokines in several organs were explored to further determine the severity of inflammation. Because of the non-inflammatory activities among the control, antibiotics alone (ATB+water) and antibiotics with PA (ATB+water+PA), the data of these groups were combined into the single "non-DSS" group. TNF-α and IL-6 were elevated in the kidneys, spleens, and hearts of ATB+DSS+PA mice, but not in ATB+DSS mice (Fig 2A and 2B). In parallel, IL-10 in ATB+DSS+PA kidney tissue was lower than in control and ATB+DSS kidney tissue, but IL-10 in ATB+DSS kidneys was not different from the control (Fig 2C). Notably, despite the increased liver enzyme and normal serum creatinine (a measure of kidney function) (Fig 1C–1E), there was inflammation in both kidneys and livers (Fig 2A–2C). For the local inflammation, the highest TNF-α and IL-6 with the lowest IL-10 in the colon tissue and the increased IL-6 with lower IL-10 in the caecum and colon of the ATB+DSS+PA mice were demonstrated (Fig 2A–2C). Meanwhile, the intestinal TNF-α, IL-6, and IL-10 were not different between ATB+DSS and control groups (Fig 2A–2C). The reduced IL-10 in the intestinal tissue (Fig 2C) together with the enhanced leaky gut of ATB+DSS+PA mice (Fig 1F–1H), supports that anti-inflammation plays a role in the prevention of gut barrier defects [14]. Due to the well-known gut-liver axis, partly through the portal veins [15, 16], endotoxemia and bacteremia might directly affect the liver in our model. Despite the non-elevated alanine transaminase and bacteremia in ATB+DSS group (Fig 1E and 1F), liver injury (histological score and hepatocyte apoptosis) was higher than in the non-DSS control group (Fig 3A–3D), possibly due to positive

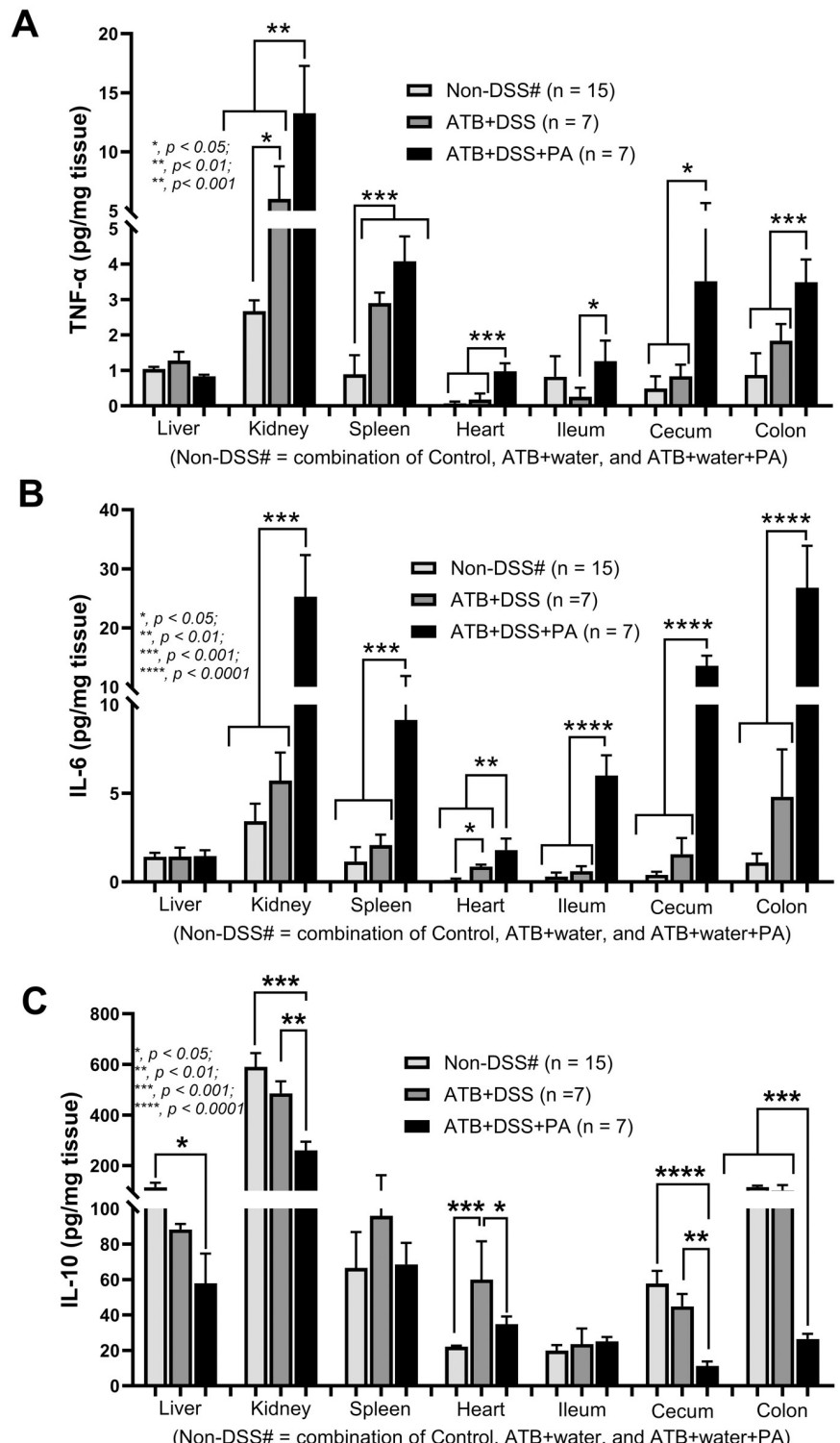

**Fig 2. *P. aeruginosa* administration induced the severity of inflammation in DSS mice with antibiotics.** Characteristics of mice in the non-DSS groups, using the combined data from water control (control), antibiotics without or with *P. aeruginosa* (ATB+water and ATB+water+PA), and the DSS groups, including dextran sulfate solution (DSS) without or with *P. aeruginosa* (ATB+DSS and ATB+DSS+PA) as indicated by cytokines (TNF-α, IL-6, and IL-10) from internal organs, including livers, kidneys, spleens, hearts, and intestines (ileum, caecum, and colon) (A-C) are demonstrated (n = 15 for the non-DSS group (Non-DSS#), and n = 7/ group for ATB+DSS and ATB+DSS

+PA). Significant differences *, $p < 0.05$; **, $p < 0.01$; ***, $p < 0.001$; ****, $p < 0.0001$ were compared between the indicated groups.

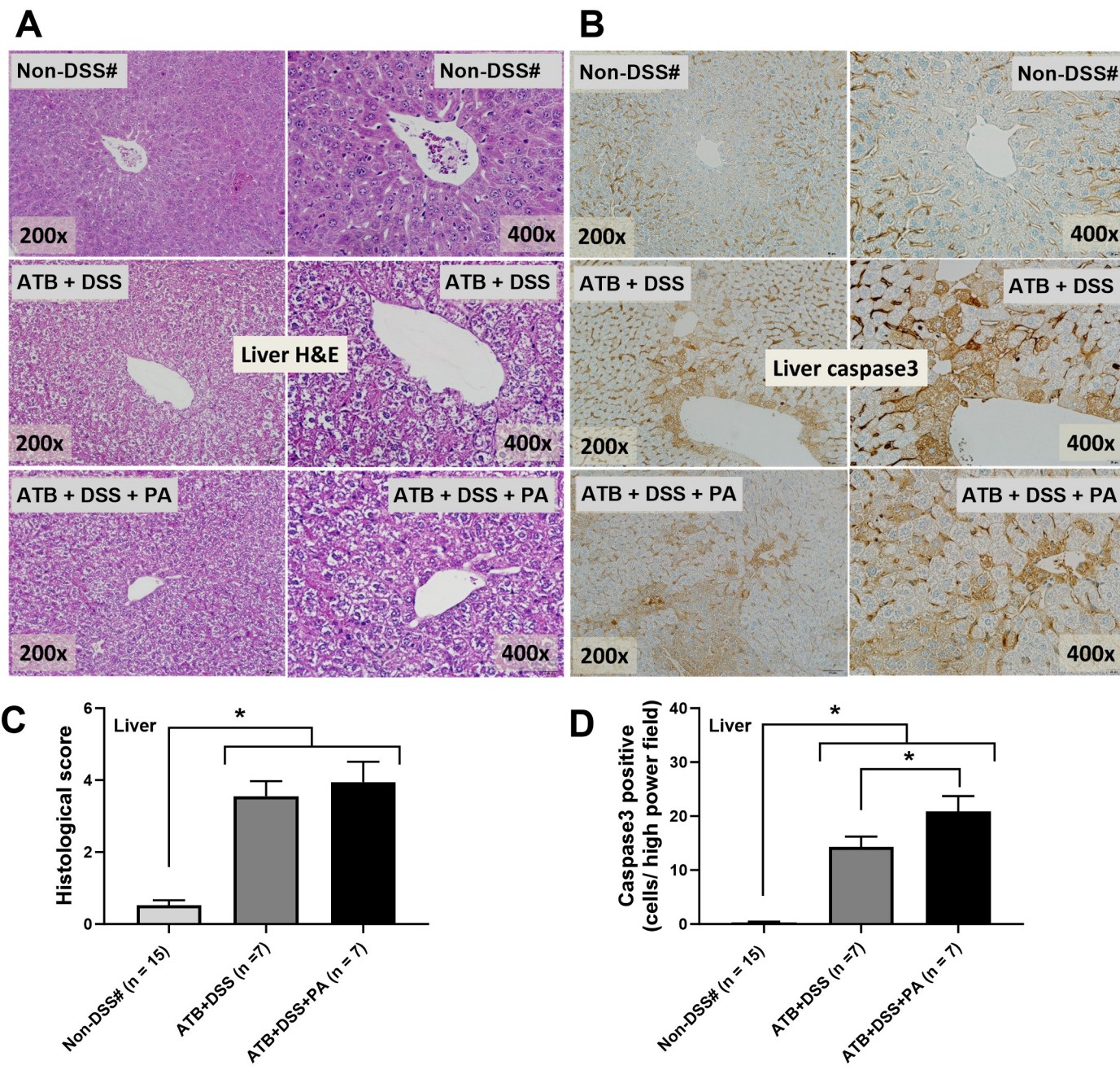

(Non-DSS# = combination of Control, ATB+water, and ATB+water+PA)

**Fig 3. *P. aeruginosa* administration worsened liver injury and apoptosis in DSS mice with antibiotics.** Characteristics of mice in the non-DSS groups, using the combined data from water control (control), antibiotics without or with *P. aeruginosa* (ATB+water and ATB+water+PA), and the DSS groups, including dextran sulfate solution (DSS) without or with *P. aeruginosa* (ATB+DSS and ATB+DSS+PA) as indicated by the representative pictures of liver histology using Hematoxylin & Eosin (H&E) staining (A) and activated caspase 3 immunohistochemistry (liver apoptosis) (B) with the histological scores and caspase 3 positive (C, D) are demonstrated (n = 15 for the non-DSS group (Non-DSS#), and n = 7/ group for ATB+DSS and ATB+DSS+PA). Significant differences **, $p < 0.01$; ****, $p < 0.0001$ were compared between the indicated groups.

endotoxemia (Fig 1H). Only liver apoptosis, not histological scoring, was higher in the ATB +DSS+PA compared to ATB+DSS group (Fig 3A–3D).

## *P. aeruginosa* administration enhanced *Pseudomonas* abundance in feces of DSS mice with antibiotics

To evaluate the impact of *P. aeruginosa* on bacterial microbiome, fecal microbiome analysis in several levels (phylum, class, order, family, genus, and the heat-map analysis on the genus) was performed (Fig 4A–4F). As such, there was a similarity between ATB+DSS and ATB+DSS+PA groups, which were distinguished from the control group by the significant difference demonstrated in Fig 5A–5D. In the phylum level of fecal microbiome analysis, when compared to the control, both ATB+DSS and ATB+DSS+PA groups showed reduced Firmicutes and Bacteroidota (the normal microbiota in healthy conditions) and increased Proteobacteria, the phylum containing the majority of Gram-negative pathogenic bacteria [14, 30, 34] (Fig 5A). In the class and order level of comparison, Gammaproteobacteria (the class of bacteria containing *Pseudomonas* spp.) and bacteria in orders Enterobacterales and Burhoderales were elevated in both ATB+DSS and ATB+DSS+PA groups compared with the control, while increased Pseudomonadales was only demonstrated in ATB+DSS+PA group (Fig 5B). At the family level, enhanced *Enterococcaceae* and *Pseudomonadaceae* were demonstrated only in ATB+DSS+PA, while similarly increased *Enterobacteriaceae* and decreased *Lachnospiraceae* were detected in both ATB+DSS and ATB+DSS+PA compared with the control (Fig 5C). In the genus level, elevated *Pseudomonas* spp. was only detected in ATB+DSS+PA mice, while *Parasutterella* spp. (Gram-negative obligate anaerobe), *Enterococcus* spp. (Gram-positive cocci causing diseases in some conditions), and *Escherichia-Shigella* spp. (Gram-negative pathogenic bacteria) were demonstrated in both ATB+DSS and ATB+DSS+PA groups (Fig 5D). Notably, the abundance of *Enterococcus* spp. and *Escherichia-Shigella* spp. in the ATB+DSS mice was lower than the ATB+DSS+PA group (Fig 5D). These data indicate an influence of *Pseudomonas* spp. administration on the presence of *Pseudomonas* spp. in mouse feces. Additionally, both ATB+DSS and ATB+DSS+PA groups had lower alpha diversity (Chao1 richness estimation and Shannon evenness estimation) than the control (Fig 6A). Indeed, the similarity was also supported by a close proximity between the DSS-administered mice that was different from the control in the Principle Coordinate Analysis (PCoA) based on Bray-Curtis dissimilarity (the dissimilarity metrics among groups represented by the distance from the axis) (Fig 6B). These data demonstrated dysbiosis (an imbalance of microbiota) in the guts of mice with ATB+DSS and ATB +DSS+PA with a subtle difference between these groups, especially in the higher abundance of *Pseudomonas* spp. in ATB+DSS+PA group.

## Lipopolysaccharide and other components from the Gram-negative bacteria induced inflammation in enterocytes (Caco-2) and hepatocytes (HepG2)

Due to i) the increased Proteobacteria and *Pseudomonas* spp. in feces (Fig 5A and 5D) with the prominent liver injury and leaky gut (FITC-dextran, bacteremia, and endotoxin) (Fig 1E–1H) in ATB+DSS+PA mice and ii) the well-known gut-liver axis through the portal veins [14, 16, 20], *Pseudomonas* components might be responsible for the enterocyte and hepatocyte injuries. Hence, the *in vitro* experiments using the heat-killed *P. aeruginosa* preparations (PA) with or without lipopolysaccharide (LPS) (representative Gram-negative bacterial molecules) were performed. In enterocytes, all stimulations (PA alone, LPS alone, and PA plus LPS) induced supernatant IL-8 from Caco-2 cells at 24 h of incubation (most prominent in PA+LPS, followed by LPS alone and PA alone), while the inflammatory genes (*IL-8*, *NF-κB*, and *TLR-2*)

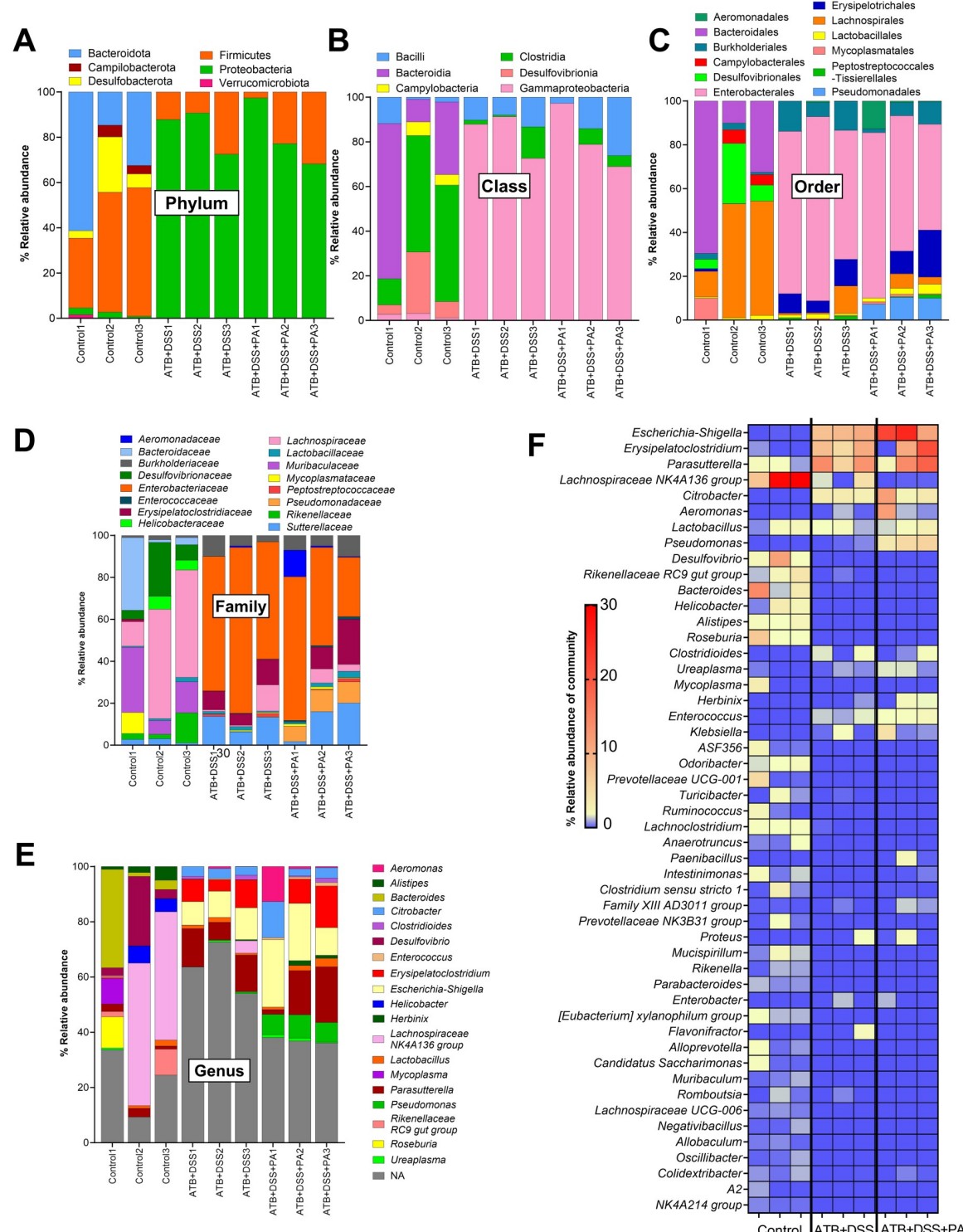

**Fig 4. *P. aeruginosa* administration enhanced bacterial dysbiosis and the number of *Pseudomonas* spp. in feces of DSS mice with antibiotics.** Characteristics of mice with drinking water (control), antibiotic plus dextran sulfate solution (ATB+DSS) with or without *P. aeruginosa* (PA) as indicated by the relative abundance of fecal bacteria in several level of the analysis (phylum, class, order, family, and genus) (A-E) together with heat-map analysis on the genus level (% relative abundance of community) (F) are demonstrated (n = 3/ group).

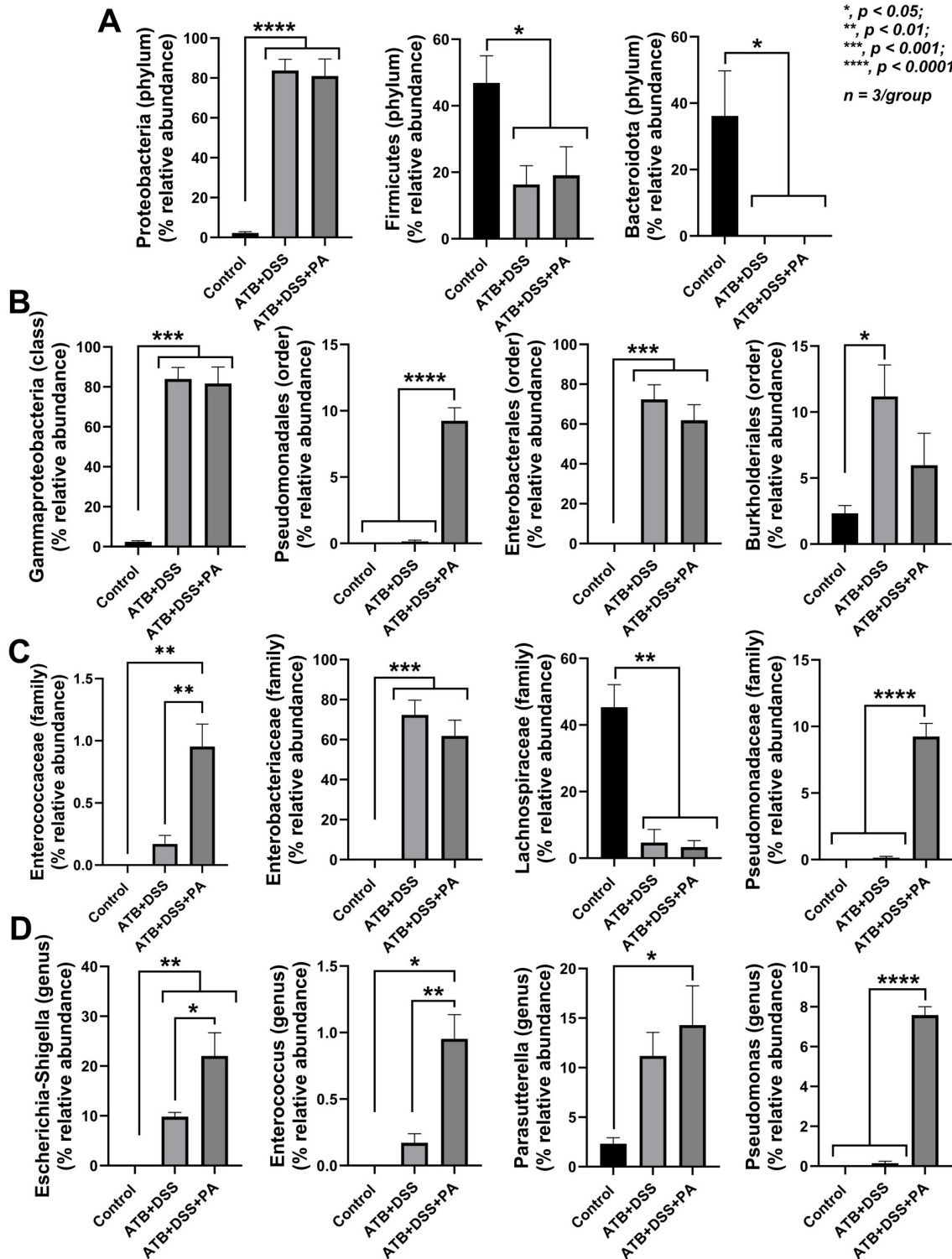

**Fig 5. *P. aeruginosa* administration altered bacterial dysbiosis and the number of *Pseudomonas* spp. in feces of DSS mice with antibiotics.** Characteristics of mice with drinking water (control), antibiotic plus dextran sulfate solution (ATB+DSS) with or without *P. aeruginosa* (PA) as indicated by the graph presentation of relative abundance of fecal bacteria in several level of the analysis (selected only the significant levels), including phylum (A), class and order (B), family (C), and genus (D) are demonstrated (n = 3/ group). Significant differences *, $p < 0.05$; **, $p < 0.01$; ***, $p < 0.001$; ****, $p < 0.0001$ were compared between the indicated groups.

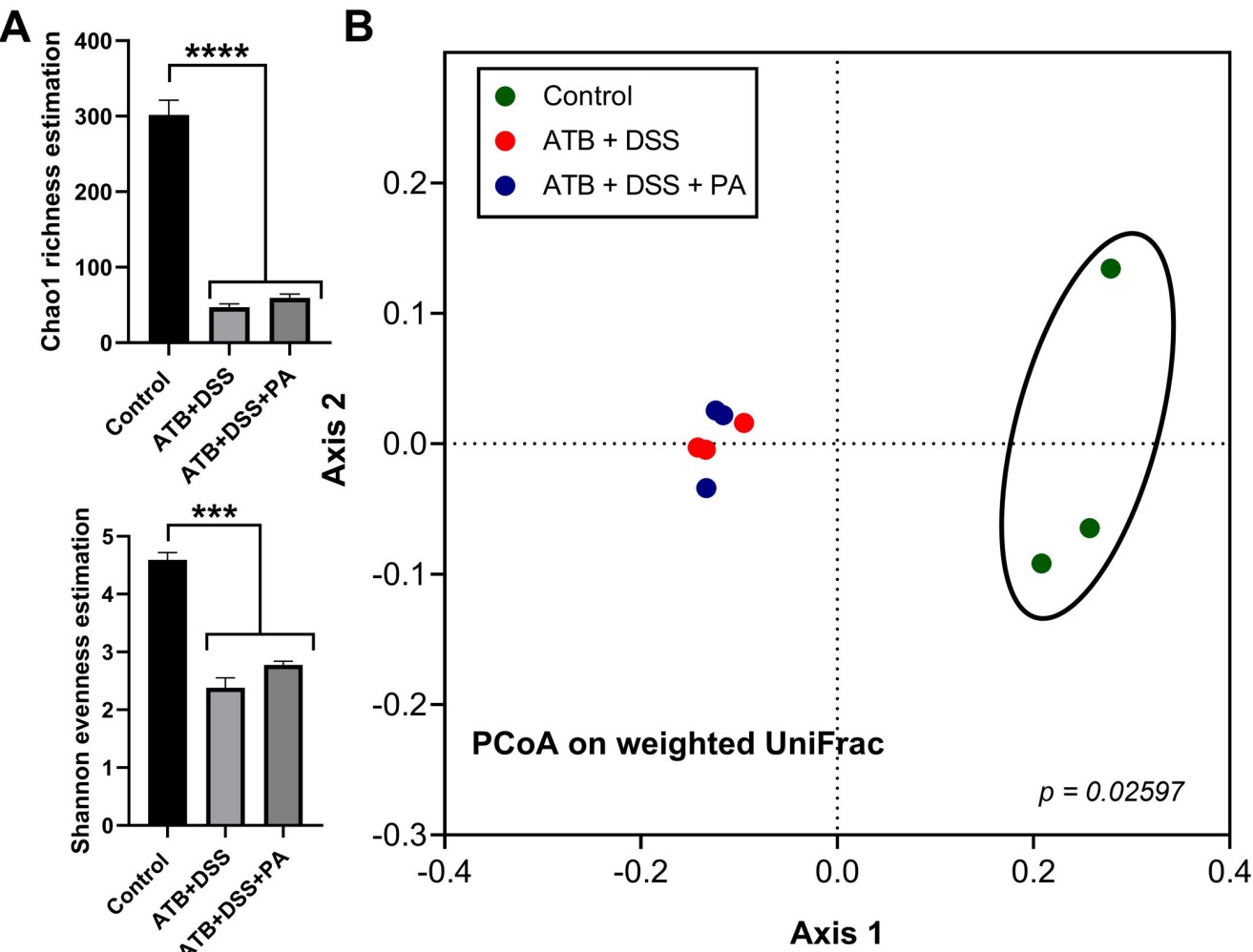

**Fig 6. The bacterial diversity in feces of *P. aeruginosa*-infected in DSS mice with antibiotics.** Characteristics of mice with drinking water (control), antibiotic plus dextran sulfate solution (ATB+DSS) with or without *P. aeruginosa* (PA) as indicated by diversity index of fecal bacteria (Chao1 richness estimation and Shannon evenness estimation) (A) with the Principle Coordinate Analysis (PCoA) based on Bray-Curtis dissimilarity (Weighted UniFrac) (B) are demonstrated (n = 3/ group). Significant differences ***, $p < 0.001$; ****, $p < 0.0001$ were compared between the indicated groups.

were most up-regulated at 2 h of incubation in PA+LPS (no up-regulation in PA alone) (Fig 7A–7D). Because peak of the cytokine production from enterocytes was at 24 h after incubation, the expression of the cytokine genes might be earlier and was performed at 2 h post-incubation [30, 34]. As such, at 2 h post-incubation, PA+LPS and LPS alone but not PA alone up-regulated *casp3* and *casp8* (apoptosis markers) but not *casp9* when compared with control (Fig 7E–7G). In parallel, PA+LPS and PA alone but not LPS alone up-regulated *NOS2* (nitric oxide synthase; a pro-inflammatory gene) (Fig 7H) which might be associated with enterocyte damage. However, there was no alteration in the expression of genes involved in mucus production (*Muc-2*) and tight junction molecules (*Claudin 1*, *occludin*, and *ZO-1*) (Fig 7I–7L).

In hepatocytes (HepG2), PA+LPS and PA alone induced higher levels of supernatant IL-8, TNF-α, and IL-10 than LPS alone at 72 h post-incubation (Fig 8A–8C). While the peak of cytokine production from the stimulated enterocytic cells was at 24 h post-incubation (Fig 7A), hepatocytic cytokine production peaked approximately at 72 h (Fig 8A–8C). Then, gene

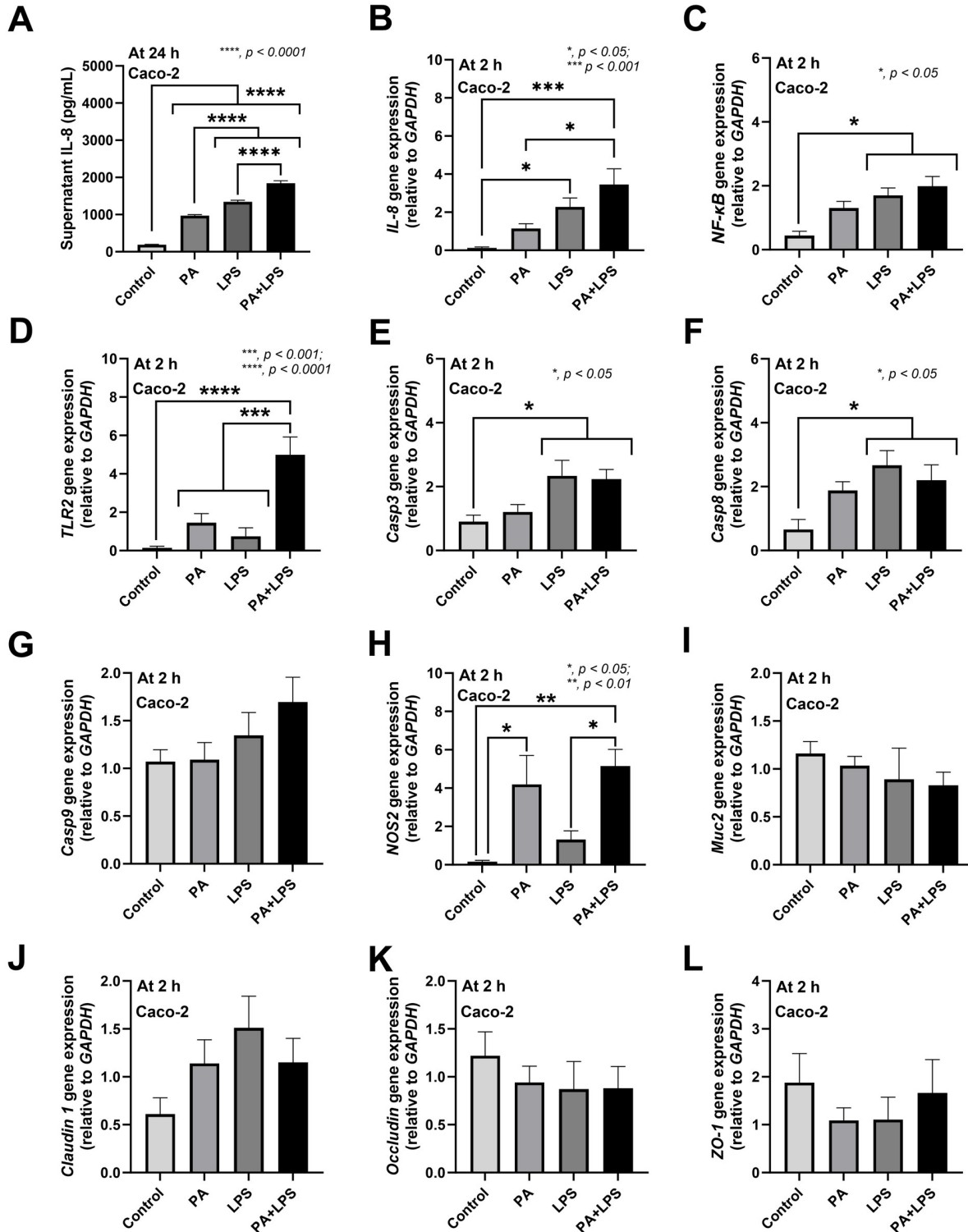

**Fig 7. Lipopolysaccharide and the *P. aeruginosa* components induced inflammation in enterocytes (Caco-2 cells).** The characteristics of enterocytes (Caco-2 cells) after activation by media control (control) and the preparations from *P. aeruginosa* alone (PA) or lipopolysaccharide (LPS) alone or in combination (PA+LPS) as indicated by supernatant IL-8 at 24 h-post activation (A) or the gene expression (2 h-post incubation) of inflammatory signals (*IL-8* and *NF-κB*, and *TLR-2*) (B-D), apoptosis markers (*casp3*, *casp8*, and *casp9*) (E-G), and enterocyte reactions, including nitric oxide synthase (*NOS2*), mucin (*Muc2*), and tight junction components (*Claudin 1*, *Occludin*, and *ZO-1*), (H-L) are demonstrated. The results were from three independent experiments each in triplicate and expressed as mean ± SEM. Significant differences *, $p < 0.05$; **, $p < 0.01$; ***, $p < 0.001$; ****, $p < 0.0001$ were compared between the indicated groups.

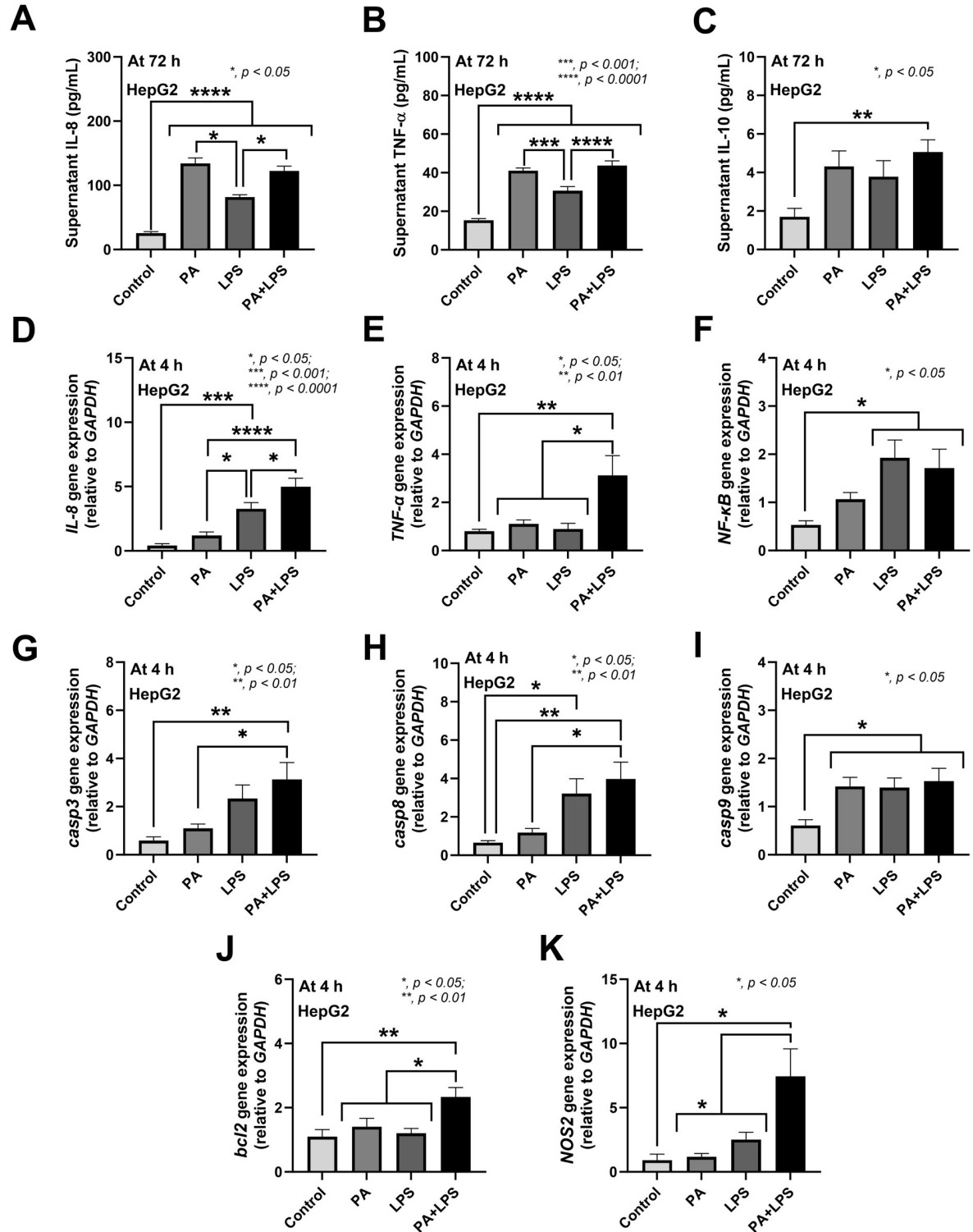

**Fig 8. Lipopolysaccharide and the *P. aeruginosa* components induced inflammation in hepatocytes (HepG2 cells).** The characteristics of hepatocytes (HepG2 cells) after activation by media control (control) and the preparations from *P. aeruginosa* alone (PA) or lipopolysaccharide (LPS) alone or in combination (PA+LPS) as indicated by supernatant cytokines IL-8, TNF-a, and IL-10 at 72 h-post activation (A-C) or the gene expression (4 h-post incubation) of inflammatory signals (*IL-8*, *TNF-α*, and *NF-κB*) (D-F), apoptosis markers (*casp3*, *casp8*, *casp9*, and *bcl2*) (G-J), and nitric oxide synthase (*NOS2*) (K) are demonstrated. The results were from three independent experiments each in triplicate and expressed as mean ± SEM. Significant differences *, $p < 0.05$; **, $p < 0.01$; ***, $p < 0.001$; ****, $p < 0.0001$ were compared between the indicated groups.

expression of cytokines in enterocytes and hepatocytes was performed at 2 and 4 h, respectively, as described in a previous publication [30]. Hence, the gene expression of other mechanisms that might be correlated with cytokines, including inflammatory process (*NOS2*), and apoptosis (*casp3*, *casp8*, *casp9* and Bcl2) was also performed at 4 h. Accordingly, at 4 h post-incubation, PA+LPS induced stronger up-regulation of inflammation genes (*IL-8*, *TNF-α*, *NF-κB*) and apoptosis (*caspase3*, *caspase8*, *bcl2*, but not *casp9*) than other groups of hepatocytes (Fig 8D–8J). Meanwhile, hepatic *NOS2* was up-regulated by PA+LPS (Fig 8K). Hence, the inflammatory synergy between *P. aeruginosa* (PA) and LPS against enterocytes and hepatocytes was possible as indicated through up-regulation in several pro-inflammatory genes.

## Discussion

*P. aeruginosa* induced gut dysbiosis that exacerbated leaky gut in mice with low dose dextran sulfate solution. In healthy mice, repeat oral administration of bacteria alone does not induce the sustained presence of that bacteria in guts and feces, partly due to the protective effect of normal microbiota and gut barrier (mucus, antimicrobial peptides, and intestinal immune responses) [52–54]. Only the repeat *P. aeruginosa* (PA) administration with microbiota interference (antibiotics) together with gut barrier damage (DSS) in the ATB+DSS+PA group induced the presence of *Pseudomonas* spp. in feces similar to a previous publication [13]. Interestingly, antibiotic-microbiota interference and mucosal injury are necessary for the model as *Pseudomonas* in feces was only detectable in ATB+DSS but not ATB+water mice (Fig 1L). With 11 days of 1.5% DSS administration, there was no overt diarrhea in any group; however, ATB+DSS+PA mice had more frequent soft stool than ATB+DSS group, supporting the possible interference of *P. aeruginosa* on the gut barrier. However, there was no mortality in all experimental groups. Normally, *P. aeruginosa* is not found in a high abundance in the intestines of mice and humans but the use of ATB, particularly with prolonged usage of antibiotics, and the defects on gut mucosa [13, 14] enhance the gut colonization of *P. aeruginosa* [12–14].

Although both ATB+DSS and ATB+DSS with *P. aeruginosa* groups induced dysbiosis, there were some notable differences between the groups. First, ATB+DSS+PA mice increased the abundance of *Pseudomonas* spp. in feces, while ATB+DSS group induced the family *Enterobacteriaceae* (the genus *Escherichia-Shigella*), the order Burkholderiales (the genus *Parasutterella*) and the family *Enterococcaceae* (the genus *Enterococcus*). Second, ATB+DSS+PA group enhanced the potentially harmful bacteria in the order Burkholderiales (the genus *Parasutterella*), *Escherichia-Shigella*, and *Enterococcus* (the potentially pathogenic bacteria in the phylum Proteobacteria), when compared with ATB+DSS group, implying a possible interaction between these bacteria. These Gram-negative aerobes might worsen the gut barrier [55] and promote the translocation of lipopolysaccharide (LPS), the predominant Gram-negative bacterial cell wall component, from the gut into the blood circulation [30] resulting in more severe inflammation [13, 14].

*P. aeruginosa* enhanced enterocyte inflammation and endotoxemia leading to liver injury in mice with low dose dextran sulfate solution. The additive effect between dysbiosis and the presence of *P. aeruginosa* in the gut might induce the more severe gut barrier defects and leaky gut-induced systemic inflammation compared with each factor alone. Gut dysbiosis in ATB+DSS group contributed to a less severe leaky gut, as indicated by endotoxemia but not bacteremia and FITC-dextran assay, when compared with the defect in all of these 3 parameters in ATB+DSS+PA mice (Fig 1F–1H). Endotoxemia without bacteremia plus less severe systemic impacts (serum cytokines, liver enzyme, and liver apoptosis) in ATB+DSS compared to ATB+DSS+PA group implies less severe gut barrier defects in the former condition. These data supported an important influence of *P. aeruginosa* in the model. Indeed, *P. aeruginosa* is a

Gram-negative bacterium that may contribute to enterocyte damage through several mechanisms, including the inflammation caused by pili, lipopolysaccharide (LPS), and polysaccharide slime (alginate), as well as the enterocyte injury caused by numerous toxins (exotoxin A, phospholipase C or hemolysin, elastase, exotoxin S, cytotoxins, and proteases) [56]. As such, some additive effects of LPS with heat-killed *P. aeruginosa* (LPS+PA) compared to LPS or PA alone in inducing pro-inflammatory enterocytes, especially IL-8 and the expression of *IL-8*, *TLR-2*, and *NOS2* (the pro-inflammatory factors), implied the possible inflammation-induced enterocyte damages. Because of the possible direct hepatic transfer of bacterial molecules from the leaky gut through the portal veins [16], hepatocytes were also tested. Similar to the enterocytes, LPS+PA demonstrated more prominent hepatic injury than LPS or PA alone through proinflammatory markers (TNF-α, IL-8, and *NOS2* expression) with increased apoptosis molecules (*caspase3*, *caspase8*, *Bcl2*). Accordingly, hepatocyte apoptosis in the livers of ATB+DSS +PA mice (endotoxemia plus bacteremia) was more prominent than in the ATB+DSS mice (endotoxemia alone). Although the liver apoptosis in ATB+DSS group (endotoxemia alone) indicated endotoxemia-induced hepatic injury, the level of LPS was insufficient to cause renal dysfunction. In comparison with non-*P. aeruginosa-* administered mice, the presence of *P. aeruginosa* in the gut of mice with DSS-induced non-diarrheal mucosal injury exacerbated prominent leaky gut that induced more severe systemic inflammation and liver damage.

Clinical aspects of the detectable *P. aeruginosa* in the healthy hosts, a clinical translation, were described. Because of the rising frequency of fecal microbiome analysis in elderly, healthy individuals, and patients with chronic diseases as a novel strategy for preventive medicine [57], reports on the discovery of *P. aeruginosa* in these hosts without gastrointestinal symptoms are increasing. However, the interpretation and further management of these individuals, especially the healthy case, with a high fecal abundance of pathogenic bacteria, especially *Pseudomonas* spp., *Klebsiella* spp., and *Salmonella* spp. are still unknown. From our results, the presence of pathogenic bacteria in feces, using *P. aeruginosa* as a representative bacterium, was as an exacerbation factor that can worsen enterocyte injury from several insults. Nevertheless, the positive fecal *P. aeruginosa* in healthy mice did not induce any symptoms as there was no leaky gut nor loose stool in ATB+water+PA mice. Thus, the further management of asymptomatic individuals with a high abundance of *P. aeruginosa* or other pathogenic bacteria in feces might be one or combined of the following choices, including further observation, leaky gut measurement, and/ or microbiota interference (probiotics or fecal transplantation). As such, the leaky gut measurement using the detection of lactulose or other gut-non-absorbable carbohydrates after an oral administration [58, 59] or probiotic use [15, 25, 60] might be beneficial for asymptomatic detection of fecal pathogenic bacteria. More studies on these topics are interesting.

## Conclusion

Impacts of *Pseudomonas aeruginosa* in the gut was tested by 5-day-administered *P. aeruginosa* in mice with antibiotics (ATB) plus a low dose dextran sulfate (DSS) (a leaky gut mouse model). The presence of *P. aeruginosa* in the gut (the elevation of Gram-negative aerobes) contributed gut dysbiosis-induced leaky gut and elevated lipopolysaccharide (LPS) levels both in the gut and in the blood circulation that directly stimulated enterocytes and hepatocytes (through the portal vein). There was gut dysbiosis in ATB with DSS mice but the addition of *P. aeruginosa* exacerbated the dysbiosis and leaky gut. While the presence of *P. aeruginosa* in the normal healthy gut was not clinically significant (possibly due to the intact gut permeability), *P. aeruginosa* presentation in the conditions with impaired gut permeability, such as ATB or DSS, resulted in more severe leaky gut and endotoxemia that further elevated systemic

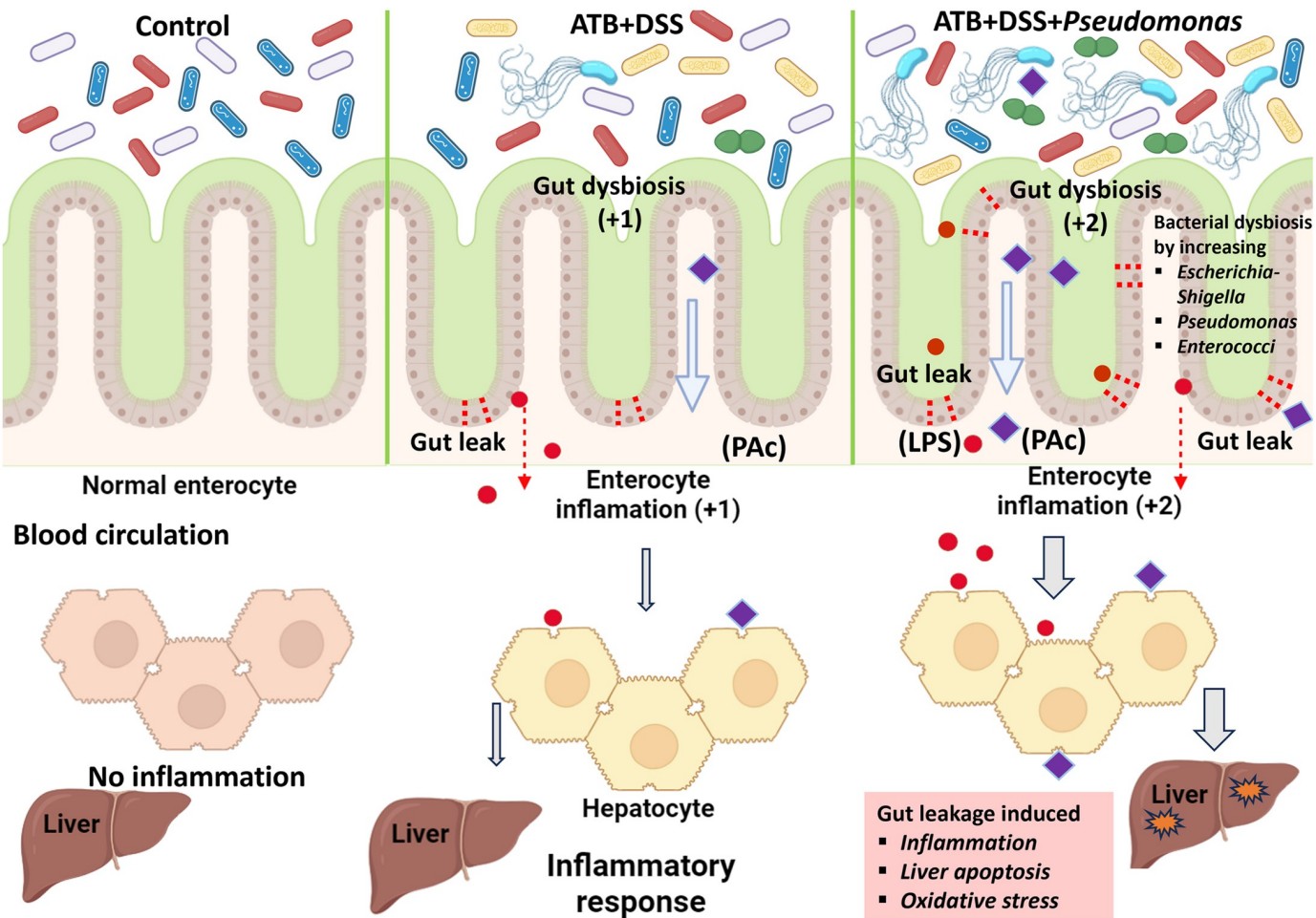

**Fig 9. The proposed working hypothesis shows the role of *P. aeruginosa* in enhancing leaky gut.** Gut dysbiosis induces leaky gut in mice treated with antibiotics and dextran sulfate solution (ATB+DSS) resulting in some degree of enterocyte inflammation (+1) and the translocation of lipopolysaccharide (LPS; red-colored circles) from the gut into the blood circulation (endotoxemia). With *P. aeruginosa*, the ATB+DSS+PA mice demonstrated i) more severe dysbiosis (+2), as indicated by some potentially pathogenic bacterial groups (*Escherichia-Shigella*, *Enterococcus*, and *Parasutterella*), ii) elevated LPS in the gut caused by *P. aeruginosa* and/or other dysbiosis-induced Gram-negative bacteria (red-colored circles in the gut lumen), iii) enhanced gut permeability damages from *P. aeruginosa* and other dysbiosis bacteria causing the direct transfer of LPS to hepatocytes (red-colored circles), partly through the portal veins that enhance systemic inflammation (inflammation, apoptosis, and increased oxidative stresses in hepatocytes), and iv) possibly elevated the translocation of intact bacteria and/or *P. aeruginosa* components (PAc) (purple-colored rectangles), as supported by higher level of bacteremia in ATB+DSS+PA than ATB+DSS mice. Hence, *P. aeruginosa* should raise concerns when it is present in the gut under situations that result in gut permeability defects, such as the use of antibiotics and colitis. Notably, the intensity of the reaction resulting from the varying abundances of bacterial molecules is shown by the thickness of the grey arrows. This picture was generated using BioRender (https://app.biorender.com/). The accessed date is 17, May, 2024 to create this picture.

inflammation, partly through LPS-induced hepatocyte responses (Fig 9). Due to the increased yearly check-up using microbiome analysis in healthy individuals, the presence of *P. aeruginosa* might have to be interpreted with the gut permeability test. More studies are interesting.

## Supporting information

**S1 Fig. *P. aeruginosa* administration exacerbated leaky gut in DSS mice with antibiotics.** Characteristics of mice in the non-DSS groups, including water control (control), antibiotics without or with *P. aeruginosa* (ATB+water and ATB+water+PA), and the DSS groups, including dextran sulfate solution (DSS) without or with *P. aeruginosa* (ATB+DSS and ATB+DSS

+PA) as indicated by colon length (A), intestinal injury score with the representative histopathology (Hematoxylin and eosin stain) (B, C). and expression of tight junction molecules (*occludin-1* and *ZO-1*) (D) are demonstrated (n = 5–7/group). Significant differences *, $p < 0.05$; **, $p < 0.01$; ***, $p < 0.001$; ****, $p < 0.0001$ were compared between the indicated groups. The representative intestinal pictures of ATB+water and ATB+water+PA are not demonstrated due to the similarity to the control group. Arrow; inflammatory cells, arrow head; inflammatory cells in group.
(TIF)

## Acknowledgments

A.L. is under the Center of Excellence on Translational Research in Inflammation and Immunology (CETRII), Department of Microbiology, Chulalongkorn University, Bangkok 10330, Thailand. We would like to thank Pornpimol Phuengmaung, Ph.D. for her assistance with photographing the morphology of the mouse colons.

## Author Contributions

**Conceptualization:** Wimonrat Panpetch, Somying Tumwasorn, Asada Leelahavanichkul.

**Data curation:** Wimonrat Panpetch, Asada Leelahavanichkul.

**Formal analysis:** Wimonrat Panpetch, Asada Leelahavanichkul.

**Funding acquisition:** Wimonrat Panpetch, Asada Leelahavanichkul.

**Investigation:** Wimonrat Panpetch.

**Methodology:** Wimonrat Panpetch, Somying Tumwasorn, Asada Leelahavanichkul.

**Project administration:** Wimonrat Panpetch, Somying Tumwasorn, Asada Leelahavanichkul.

**Resources:** Wimonrat Panpetch, Somying Tumwasorn, Asada Leelahavanichkul.

**Software:** Wimonrat Panpetch, Asada Leelahavanichkul.

**Supervision:** Somying Tumwasorn, Asada Leelahavanichkul.

**Validation:** Wimonrat Panpetch, Asada Leelahavanichkul.

**Visualization:** Wimonrat Panpetch, Asada Leelahavanichkul.

**Writing – original draft:** Wimonrat Panpetch, Asada Leelahavanichkul.

**Writing – review & editing:** Wimonrat Panpetch, Somying Tumwasorn, Asada Leelahavanichkul.

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
