## [Decision Letter · Decision Letter 0]

15 May 2024

PONE-D-24-13045Presence of Pseudomonas aeruginosa in feces exacerbate leaky gut in mice with low dose dextran sulfate solution, impacts of specific bacteriaPLOS ONE

Dear Dr. Leelahavanichkul,

Thank you for submitting your manuscript to PLOS ONE. After careful consideration, we feel that it has merit but does not fully meet PLOS ONE’s publication criteria as it currently stands. Therefore, we invite you to submit a revised version of the manuscript that addresses the points raised during the review process. It was felt that the authors need to provide the data pertaining to at least the expression of genes for the tight junction proteins.  In addition, a rationale for the use of 2h time point for the effect on enterocytes, whereas 4h time point used for the gene expression profile for the hepatocytes should be provided.  Also, several details including the  type of LPS was missing.  Please submit the responses to each of the reviewer's concerns. Please submit your revised manuscript by Jun 29 2024 11:59PM. If you will need more time than this to complete your revisions, please reply to this message or contact the journal office at plosone@plos.org. Please include the following items when submitting your revised manuscript:A rebuttal letter that responds to each point raised by the academic editor and reviewer(s). You should upload this letter as a separate file labeled 'Response to Reviewers'.A marked-up copy of your manuscript that highlights changes made to the original version. You should upload this as a separate file labeled 'Revised Manuscript with Track Changes'.An unmarked version of your revised paper without tracked changes. You should upload this as a separate file labeled 'Manuscript'.

We look forward to receiving your revised manuscript.

Kind regards,

Pradeep Dudeja

Academic Editor

PLOS ONE

Journal Requirements:

   "This research is supported by the Program Management Unit for Human Resources & Institutional Development, Research and Innovation (B16F640175) with Rachadapisek Sompote Matching Fund (RA-MF-22/65 and RA-MF-13/66), and Rachadapisek Sompote Endowment Fund (RA66/008 and RA66/009), as well as National Research Council of Thailand (NRCT-N41A640076 and NRCT-N34A660583). WP was supported by Rachadapisek Sompote Fund for Postdoctoral Fellowship, Chulalongkorn University. " 

Reviewers' comments:

Reviewer's Responses to Questions

**Comments to the Author**

1. Is the manuscript technically sound, and do the data support the conclusions?

Reviewer #1: Yes

Reviewer #2: Yes

2. Has the statistical analysis been performed appropriately and rigorously? 

Reviewer #1: No

Reviewer #2: Yes

3. Have the authors made all data underlying the findings in their manuscript fully available?

Reviewer #1: Yes

Reviewer #2: Yes

4. Is the manuscript presented in an intelligible fashion and written in standard English?

Reviewer #1: Yes

Reviewer #2: Yes

5. Review Comments to the Author

Reviewer #1: This study demonstrates that the presence of pseudomonas aeruginosa (PA) in the gut exacerbated DSS-induced intestinal injury with spontaneous translocation of LPS and bacteria from the gut into the blood circulation that induced severe systemic inflammation using mice as an in vivo model system. Although the study is interesting, there are several issues that need to be addressed.

1. Statistical analysis must be done for the entire data presented in this manuscript. Most of the p-values presented appears to be wrong.

2. Provide colon length and morphology of controls and ATB+PA+DSS treated mice.

3. Specify the type of LPS used in this study.

4. Provide the tight junction proteins gene expression levels in controls and ATB+PA+DSS treated mice colon.

5. Many typographical and grammatical errors throughout the manuscript.

Reviewer #2: In this interesting study, the authors tested the hypothesis that increased abundance of pathogenic bacteria (specifically of the phylum Proteobacteria) in healthy hosts may increase of the healthy hosts their susceptibility to intestinal injury. The authors investigated the impact of Pseudomonas aeruginosa (PA, the representative Proteobacteria) in a mouse model with non-diarrheal gut permeability defect using 1.5% dextran sulfate solution (DSS) plus antibiotics (ATB) with or without orally administered PA. The results showed that ATB+DSS+PA induced more severe intestinal injury as indicated by stool consistency and leaky gut (FITC dextran assay, bacteremia, and endotoxemia) with an increase in serum cytokines, liver enzyme, and hepatocyte apoptosis when compared with ATB+DSS mice. However, these parameters remained unchanged in the non-DSS group: water alone (Control), antibiotics alone (ATB+water), and antibiotics with PA (ATB+water+PA). A similar fecal microbiome pattern was observed between ATB+DSS and ATB+DSS+PA. Moreover, a higher abundance of Pseudomonas, Enterococci, and Escherichia-Shigella was detected in ATB+DSS+PA compared to ATB+DSS. Additionally, heat-killed P. aeruginosa (PA) and LPS together induced enterocyte (Caco2 cells) injury and hepatic injury compared to LPS or PA alone via increased expression of proinflammatory markers (TNF-α, IL-8, and NOS2) and apoptosis proteins (caspase3, caspase8, Bcl2). Based on these data, the authors concluded that the presence of P. aeruginosa in the gut of mice with DSS-induced non-diarrheal mucosal injury exacerbated prominent leaky gut that induced more severe systemic inflammation.

This study has clinical relevance as it can provide insights on how to interpret and further manage healthy individuals with a high fecal abundance of pathogenic bacteria, especially Pseudomonas spp., Klebsiella spp., and Salmonella spp. that still remains a challenge. The abstract is concise, and the conclusions are well supported by the relevant data. However, the authors should consider the following suggestions:

1. Did the authors examine the expression level of tight junction proteins such as occludin, claudins 1, 2, 4 and ZO1 in ATB+DSS+PA mice compared to ATB+DSS mice? The authors should provide a compelling rationale as to why both mice were used for the current study.

2. Any specific reason as to why 2h (enterocytes) and 4h (hepatocytes) time points were chosen to examine the effects of PA + LPS on the expression various genes including IL8, TNF-α, NOS2, caspases 3, 8, 9 and Bcl2?

3. Figure legends 1-3 (lines 70, 709, 717)- should be ATB + water + PA instead of ATB + ATB + PA

4. Figure legend 7 (line 741)- should be 2h instead of 4h

5. Line 288- It should be colon not ileum

6. The authors should proof-read the manuscript carefully to avoid typo and grammatical errors.

6. PLOS authors have the option to publish the peer review history of their article (what does this mean?). If published, this will include your full peer review and any attached files.

Reviewer #1: No

Reviewer #2: No

---

## [Author Response · Author response to Decision Letter 0]

28 May 2024

PONE-D-24-13045

Presence of Pseudomonas aeruginosa in feces exacerbate leaky gut in mice with low dose dextran sulfate solution, impacts of specific bacteria

PLOS ONE

Dear Dr. Leelahavanichkul,

Thank you for submitting your manuscript to PLOS ONE. After careful consideration, we feel that it has merit but does not fully meet PLOS ONE’s publication criteria as it currently stands. Therefore, we invite you to submit a revised version of the manuscript that addresses the points raised during the review process.

It was felt that the authors need to provide the data pertaining to at least the expression of genes for the tight junction proteins. In addition, a rationale for the use of 2h time point for the effect on enterocytes, whereas 4h time point used for the gene expression profile for the hepatocytes should be provided. Also, several details including the type of LPS was missing. Please submit the responses to each of the reviewer's concerns.

ANS: We thank the editor for the comment and we provide the tight junction proteins gene expression in colon of each group of mice. Moreover, we add the discussion on the result section about a rationale for the use of 2h time point for the effect on enterocytes, whereas 4h time point used for the gene expression profile for the hepatocytes at the lines 443-445 and 467-473.

We look forward to receiving your revised manuscript.

Kind regards,

Pradeep Dudeja

Academic Editor

PLOS ONE

Journal Requirements:

ANS: We thank the editor for the comment and we ensure that our manuscript meets PLOS one’ s style requirements.

ANS: We remove funding from the revised manuscript.

 "This research is supported by the Program Management Unit for Human Resources & Institutional Development, Research and Innovation (B16F640175) with Rachadapisek Sompote Matching Fund (RA-MF-22/65 and RA-MF-13/66), and Rachadapisek Sompote Endowment Fund (RA66/008 and RA66/009), as well as National Research Council of Thailand (NRCT-N41A640076 and NRCT-N34A660583). WP was supported by Rachadapisek Sompote Fund for Postdoctoral Fellowship, Chulalongkorn University. " 

ANS: We amend the role of funders “The funders had no role in study design, data collection and analysis, decision to publish, or preparation of the manuscript." 

ANS: We thank the editor for the comment and we provide the minimal data set for publication in the excel file.

ANS: All data were included in the minimal data set for publication. If you require additional information, please contact the corresponding author.

Reviewers' comments:

Reviewer's Responses to Questions

Comments to the Author

1. Is the manuscript technically sound, and do the data support the conclusions?

Reviewer #1: Yes

Reviewer #2: Yes

2. Has the statistical analysis been performed appropriately and rigorously?

Reviewer #1: No

Reviewer #2: Yes

3. Have the authors made all data underlying the findings in their manuscript fully available?

Reviewer #1: Yes

Reviewer #2: Yes

4. Is the manuscript presented in an intelligible fashion and written in standard English?

Reviewer #1: Yes

Reviewer #2: Yes

5. Review Comments to the Author

Reviewer #1: This study demonstrates that the presence of pseudomonas aeruginosa (PA) in the gut exacerbated DSS-induced intestinal injury with spontaneous translocation of LPS and bacteria from the gut into the blood circulation that induced severe systemic inflammation using mice as an in vivo model system. Although the study is interesting, there are several issues that need to be addressed.

1. Statistical analysis must be done for the entire data presented in this manuscript. Most of the p-values presented appears to be wrong.

ANS: We thank the reviewer for the comment and we check, correct all statistical analysis in all figures. The p-values have been changed.

2. Provide colon length and morphology of controls and ATB+PA+DSS treated mice.

ANS: We thank the reviewer for the comment and we provide colon length and morphology of controls and ATB+PA+DSS treated mice in supplement data (S1 Fig).

3. Specify the type of LPS used in this study.

ANS: We thank the reviewer for the comment and we add the type of LPS in revised manuscript in the method section “lipopolysaccharide (LPS) from Escherichia coli O26:B6 (Sigma-Aldrich) at 1 µg/mL” at the lines 263-264.

4. Provide the tight junction proteins gene expression levels in controls and ATB+PA+DSS treated mice colon.

ANS: We thank the reviewer for the comment and we provide the tight junction proteins gene expression level in controls and ATB+PA+DSS treated mice colon in the supplement data at the lines 310-311. 

5. Many typographical and grammatical errors throughout the manuscript.

ANS: We thank the reviewer for the comment and we check and correct the typographical and grammatical errors throughout the revised manuscript and labeled with yellow highlight.

Reviewer #2: In this interesting study, the authors tested the hypothesis that increased abundance of pathogenic bacteria (specifically of the phylum Proteobacteria) in healthy hosts may increase of the healthy hosts their susceptibility to intestinal injury. The authors investigated the impact of Pseudomonas aeruginosa (PA, the representative Proteobacteria) in a mouse model with non-diarrheal gut permeability defect using 1.5% dextran sulfate solution (DSS) plus antibiotics (ATB) with or without orally administered PA. The results showed that ATB+DSS+PA induced more severe intestinal injury as indicated by stool consistency and leaky gut (FITC dextran assay, bacteremia, and endotoxemia) with an increase in serum cytokines, liver enzyme, and hepatocyte apoptosis when compared with ATB+DSS mice. However, these parameters remained unchanged in the non-DSS group: water alone (Control), antibiotics alone (ATB+water), and antibiotics with PA (ATB+water+PA). A similar fecal microbiome pattern was observed between ATB+DSS and ATB+DSS+PA. Moreover, a higher abundance of Pseudomonas, Enterococci, and Escherichia-Shigella was detected in ATB+DSS+PA compared to ATB+DSS. Additionally, heat-killed P. aeruginosa (PA) and LPS together induced enterocyte (Caco2 cells) injury and hepatic injury compared to LPS or PA alone via increased expression of proinflammatory markers (TNF-α, IL-8, and NOS2) and apoptosis proteins (caspase3, caspase8, Bcl2). Based on these data, the authors concluded that the presence of P. aeruginosa in the gut of mice with DSS-induced non-diarrheal mucosal injury exacerbated prominent leaky gut that induced more severe systemic inflammation.

This study has clinical relevance as it can provide insights on how to interpret and further manage healthy individuals with a high fecal abundance of pathogenic bacteria, especially Pseudomonas spp., Klebsiella spp., and Salmonella spp. that still remains a challenge. The abstract is concise, and the conclusions are well supported by the relevant data. However, the authors should consider the following suggestions:

1. Did the authors examine the expression level of tight junction proteins such as occludin, claudins 1, 2, 4 and ZO1 in ATB+DSS+PA mice compared to ATB+DSS mice? 

The authors should provide a compelling rationale as to why both mice were used for the current study.

ANS: We thank the reviewer for the comment and we provide the tight junction proteins gene expression in colon of each group of mice at the lines 310-311 (S1 Fig). 

We add the rationale why both mice were used for this study in the Fig. 9 and in conclusion section.

2. Any specific reason as to why 2h (enterocytes) and 4h (hepatocytes) time points were chosen to examine the effects of PA + LPS on the expression various genes including IL8, TNF-α, NOS2, caspases 3, 8, 9 and Bcl2?

ANS: We thank the reviewer for the comment and we answer the specific reason in this point. 

Because peak of the cytokine production from enterocytes was at 24 h after incubation, the expression of the cytokine genes might be earlier and was performed at 2 h post-incubation [30, 34] at the lines 443-445. 

While the peak of cytokine production from the stimulated enterocytic cells was at 24 h post-incubation (Fig 7A), hepatocytic cytokine production peaked approximately at 72 h (Figs 8A-C). Then, gene expression of cytokines in enterocytes and hepatocytes was performed at 2 and 4 h, respectively, as described in a previous publication [30]. Hence, the gene expression of other mechanisms that might be correlated with cytokines, including inflammatory process (NOS2), and apoptosis (casp3, casp8, casp9 and Bcl2) was also performed at 4 h at the lines 467-473.

3. Figure legends 1-3 (lines 70, 709, 717)- should be ATB + water + PA instead of ATB + ATB + PA

ANS: We thank the reviewer for the comment and we correct accordingly.

4. Figure legend 7 (line 741)- should be 2h inste

---

## [Decision Letter · Decision Letter 1]

19 Jul 2024

PONE-D-24-13045R1Presence of Pseudomonas aeruginosa in feces exacerbate leaky gut in mice with low dose dextran sulfate solution, impacts of specific bacteriaPLOS ONE

Dear Dr. Leelahavanichkul,

Thank you for submitting your manuscript to PLOS ONE. After careful consideration, we feel that your revised manuscript has significantly improved.  However, one of the reviewer has some minor concerns.  We request you to please address those concerns and submit the revised version as soon as possible.

We look forward to receiving your revised manuscript.

Kind regards,

Pradeep Dudeja

Academic Editor

PLOS ONE

Journal Requirements:

Reviewers' comments:

Reviewer's Responses to Questions

**Comments to the Author**

1. If the authors have adequately addressed your comments raised in a previous round of review and you feel that this manuscript is now acceptable for publication, you may indicate that here to bypass the “Comments to the Author” section, enter your conflict of interest statement in the “Confidential to Editor” section, and submit your "Accept" recommendation.

Reviewer #1: All comments have been addressed

Reviewer #2: All comments have been addressed

2. Is the manuscript technically sound, and do the data support the conclusions?

Reviewer #1: Yes

Reviewer #2: Yes

3. Has the statistical analysis been performed appropriately and rigorously? 

Reviewer #1: Yes

Reviewer #2: Yes

4. Have the authors made all data underlying the findings in their manuscript fully available?

Reviewer #1: Yes

Reviewer #2: Yes

5. Is the manuscript presented in an intelligible fashion and written in standard English?

Reviewer #1: Yes

Reviewer #2: Yes

6. Review Comments to the Author

Reviewer #1: Authors addressed all of my concerns/suggestions raised during my first review which includes additional data.

Reviewer #2: The authors have duly addressed the concerns raised by the reviewers. However, they should consider the following minor suggestions:

1. Line 311- It is 'Occludin" not 'Occluding'

2. For the sake of clarity and to avoid confusion, Legends to Figures 1-8 and 9 (proposed model) should be mentioned separately under "Figure Legends' (after 'References' and Table 1).

7. PLOS authors have the option to publish the peer review history of their article (what does this mean?). If published, this will include your full peer review and any attached files.

Reviewer #1: No

Reviewer #2: No

---

## [Author Response · Author response to Decision Letter 1]

20 Jul 2024

PONE-D-24-13045R1

Presence of Pseudomonas aeruginosa in feces exacerbate leaky gut in mice with low dose dextran sulfate solution, impacts of specific bacteria

PLOS ONE

Dear Dr. Leelahavanichkul,

Thank you for submitting your manuscript to PLOS ONE. After careful consideration, we feel that your revised manuscript has significantly improved. However, one of the reviewer has some minor concerns. We request you to please address those concerns and submit the revised version as soon as possible.

ANS: We change the financial disclosure and update statement in the cover letter.

We look forward to receiving your revised manuscript.

Kind regards,

Pradeep Dudeja

Academic Editor

PLOS ONE

Journal Requirements:

ANS: We thank the editor for the comment and we remove reference and replace a relevant current reference. 

Reviewers' comments:

Reviewer's Responses to Questions

Comments to the Author

1. If the authors have adequately addressed your comments raised in a previous round of review and you feel that this manuscript is now acceptable for publication, you may indicate that here to bypass the “Comments to the Author” section, enter your conflict of interest statement in the “Confidential to Editor” section, and submit your "Accept" recommendation.

Reviewer #1: All comments have been addressed

Reviewer #2: All comments have been addressed

2. Is the manuscript technically sound, and do the data support the conclusions?

Reviewer #1: Yes

Reviewer #2: Yes

3. Has the statistical analysis been performed appropriately and rigorously?

Reviewer #1: Yes

Reviewer #2: Yes

4. Have the authors made all data underlying the findings in their manuscript fully available?

Reviewer #1: Yes

Reviewer #2: Yes

5. Is the manuscript presented in an intelligible fashion and written in standard English?

Reviewer #1: Yes

Reviewer #2: Yes

6. Review Comments to the Author

Reviewer #1: Authors addressed all of my concerns/suggestions raised during my first review which includes additional data.

Reviewer #2: The authors have duly addressed the concerns raised by the reviewers. However, they should consider the following minor suggestions:

1. Line 311- It is 'Occludin" not 'Occluding'

ANS: We thank the reviewer for the comment and we correct accordingly.

2. For the sake of clarity and to avoid confusion, Legends to Figures 1-8 and 9 (proposed model) should be mentioned separately under "Figure Legends' (after 'References' and Table 1).

ANS: We thank the reviewer for the comment and we correct following reviewer comment.

7. PLOS authors have the option to publish the peer review history of their article (what does this mean?). If published, this will include your full peer review and any attached files.

Do you want your identity to be public for this peer review? For information about this choice, including consent withdrawal, please see our Privacy Policy.

Reviewer #1: No

Reviewer #2: No

While revising your submission, please upload your figure files to the Preflight Analysis and Conversion Engine (PACE) digital diagnostic tool, https://pacev2.apexcovantage.com/. PACE helps ensure that figures meet PLOS requirements. To use PACE, you must first register as a user. Registration is free. Then, login and navigate to the UPLOAD tab, where you will find detailed instructions on how to use the tool. If you encounter any issues or have any questions when using PACE, please email PLOS at <a href="mailto:figures@plos.org">figures@plos.org. Please note that Supporting Information files do not need this step.

---

## [Editor Report · Decision Letter 2]

6 Aug 2024

Presence of Pseudomonas aeruginosa in feces exacerbate leaky gut in mice with low dose dextran sulfate solution, impacts of specific bacteria

PONE-D-24-13045R2

Dear Dr. Leelahavanichkul,

We’re pleased to inform you that your manuscript has been judged scientifically suitable for publication and will be formally accepted for publication once it meets all outstanding technical requirements.

Kind regards,

Pradeep Dudeja

Academic Editor

PLOS ONE
---

## [Editor Report · Acceptance letter]

17 Sep 2024

PONE-D-24-13045R2 

PLOS ONE

Dear Dr. Leelahavanichkul, 

I'm pleased to inform you that your manuscript has been deemed suitable for publication in PLOS ONE. Congratulations! Your manuscript is now being handed over to our production team.

Kind regards, 

on behalf of

Dr. Pradeep Dudeja 

Academic Editor

PLOS ONE